# E-Skin and Its Advanced Applications in Ubiquitous Health Monitoring

**DOI:** 10.3390/biomedicines12102307

**Published:** 2024-10-11

**Authors:** Xidi Sun, Xin Guo, Jiansong Gao, Jing Wu, Fengchang Huang, Jia-Han Zhang, Fuhua Huang, Xiao Lu, Yi Shi, Lijia Pan

**Affiliations:** 1Collaborative Innovation Center of Advanced Microstructures, School of Electronic Science and Engineering, Nanjing University, Nanjing 210093, China; dg21230047@smail.nju.edu.cn (X.S.); dg21230017@smail.nju.edu.cn (X.G.); gaojiansong1209@163.com (J.G.); jing-wu@smail.nju.edu.cn (J.W.); fengchang4444@126.com (F.H.); 2School of Electronic Information Engineering, Inner Mongolia University, Hohhot 010021, China; jiahan_zhang@outlook.com; 3Department of Thoracic and Cardiovascular Surgery, Nanjing First Hospital, Nanjing Medical University, Nanjing 210006, China; huangfuhua@sina.cn; 4The First Affiliated Hospital of Nanjing Medical University, Jiangsu Province Hospital, Nanjing 210093, China; luxiao1972@163.com

**Keywords:** flexible electronics, e-skin, health monitoring, sensors, wireless

## Abstract

E-skin is a bionic device with flexible and intelligent sensing ability that can mimic the touch, temperature, pressure, and other sensing functions of human skin. Because of its flexibility, breathability, biocompatibility, and other characteristics, it is widely used in health management, personalized medicine, disease prevention, and other pan-health fields. With the proposal of new sensing principles, the development of advanced functional materials, the development of microfabrication technology, and the integration of artificial intelligence and algorithms, e-skin has developed rapidly. This paper focuses on the characteristics, fundamentals, new principles, key technologies, and their specific applications in health management, exercise monitoring, emotion and heart monitoring, etc. that advanced e-skin needs to have in the healthcare field. In addition, its significance in infant and child care, elderly care, and assistive devices for the disabled is analyzed. Finally, the current challenges and future directions of the field are discussed. It is expected that this review will generate great interest and inspiration for the development and improvement of novel e-skins and advanced health monitoring systems.

## 1. Introduction

Skin, the largest organ of the human body, covers the surface of the body and not only has physical properties such as breathability, moisture conductivity, flexibility, and elasticity but also functional properties such as hazard shielding, self-repairing, force sensing, temperature sensing, and humidity sensing. In other words, it serves as an effective protective barrier, shielding the human body from external injuries while providing an ideal platform for perceiving and responding to external stimuli. Electronic skin (e-skin) is a bionic electronic system that mimics the function of human skin, which is usually made of flexible and stretchable materials with embedded sensors for sensing the external environment (e.g., pressure, temperature, humidity, chemicals, etc.) as well as for monitoring the human body’s physiological signals (e.g., heart rate, blood oxygenation, muscular activity, etc.). Due to its flexibility, stretchability, and conformability, e-skin can accomplish continuous, real-time, non-sensory, and non-invasive biosignal detection that is not possible with traditional clunky medical devices. The concept of e-skin dates back to the 1970s, when research focused on basic flexible electronics and attempts were made to develop flexible sensor systems with simple tactile sensing capabilities, and these early prototypes laid the groundwork for the later development of e-skin. With rapid advances in nanotechnology, flexible electronics, and materials science, e-skins have achieved significant breakthroughs in sensing functionality, flexibility, and integration. Researchers have developed electronic skins with multimodal sensing capabilities, capable of simultaneously detecting multiple parameters such as pressure, temperature, humidity, and chemical environment. Additionally, developments in flexible circuits and stretchable materials have enabled e-skin to adhere more closely to human skin, enhancing its potential application in medical and health monitoring. In recent years, as technology has shown a burst of progress, the fourth industrial revolution, which is dominated by artificial intelligence technology, digitalization technology, and the Internet of Things (IoT), has erupted [1]. E-skins with intelligent sensing and interaction capabilities offer unprecedented possibilities for the development of personal protection, healthcare, human–computer interaction, and smart manufacturing, among other fields. Furthermore, e-skin has certain protective capabilities, sub environmental energy harvesting and conversion capabilities, and self-repairing capabilities to better mimic skin functions for personal protection, self-driven sensing, and reliable sensing [2]. Thus, e-skin shows great potential for applications in real-time physiological signal monitoring, health conditions, and personalized medicine. Simultaneously, providing seamless comfort in daily use and ensuring long-term stable physiological signal monitoring have become key focuses in the ongoing development of e-skin.

To ensure compatibility with nonsensory comfort and long-term stability, the e-skin needs to have several characteristics. First, the core is the sensor, and excellent and stable sensing performance is the basis for ensuring that the e-skin can be stable and accurate. Various artificial microstructures can be constructed to improve the optical, electrical, force, and thermal properties of e-skin [3,4,5]. Simultaneously, exploring new mechanisms to improve the performance of e-skin, extending its functions, and deepening its applications are the new directions to ensure the comfortable and stable monitoring of the physiological signals [6]. Second, the development and design of materials with attributes such as breathability, ultra-thinness, conformality, and stretchability are crucial to optimizing the comfort and user experience of e-skin (Figure 1) [7,8,9,10]. Currently, achieving these features in a single material is key to improving the comfort and usability of e-skin. Although materials possessing one or more of these properties have been developed, creating devices that integrate a combination of these features remains a critical research direction for ensuring long-term comfort during e-skin wear. Finally, independent operating frame design and wireless signal transmission technologies are vital for significantly enhancing e-skin performance and translating laboratory prototypes into practical applications. Although the front-end data acquisition part of the traditional e-skin has a high index of reference performance in the laboratory, the dependence on the test environment and graceful data acquisition greatly hinder the application in the actual health monitoring. Currently, e-skin data analysis usually relies on manual supervision in the lab, and the later signal selection, evaluation, and processing cannot be performed in real time, which seriously affects the timeliness and convenience of health monitoring. Combined with today’s rapidly evolving artificial intelligence, the emerging e-skin allows for data analysis in decoding large, complex maps generated from a variety of signals by continuously monitoring multimodal data [11,12]. Deep learning can uncover medical insights that are challenging to achieve through traditional methods while providing accurate predictions that can mimic or even surpass human expertise [13,14,15]. However, achieving an e-skin system with comfortable wearability, long-term stable physiological signal monitoring, and real-time accurate health analysis and disease prediction will require significant research and development efforts over an extended period.

While previous reports have thoroughly covered e-skin’s material selection, synthesis methods, and signal acquisition and transmission, the rapid technological advancements demand an update. In the face of emerging materials, innovative synthesis techniques, advanced micromachining technologies, and the expanding applications in healthcare, it is crucial to present the latest findings. This review aims to synthesize recent progress and address current challenges in applying e-skin to healthcare. Firstly, a discussion is made around the required performance of e-skin applied to health monitoring. Next, we analyze the basic sensing principles and technical components of an e-skin, including accurate signal acquisition, wireless transmission, intelligent analysis, and energy harvesting technologies. This is followed by a comprehensive description of the applications of e-skin in recent years in the fields of health management, medical monitoring, and integrated system monitoring for different populations. Finally, the current challenges and future development directions of e-skin in healthcare are discussed.

## 2. Important Features of E-Skin for Health Monitoring

The e-skin is a flexible electronic device that mimics the human skin, so it needs to mimic the basic property of human skin, flexibility, first and foremost. In addition, in order to meet the requirements of comfortable wearing and stable sensing performance in different environments, the e-skin should also possess the properties of breathability and conformability [4,16,17,18]. Of course, as an e-skin incorporating advanced technologies and materials, the human-safe and environmentally friendly biodegradability is also extremely important. In this section, we discuss these important properties in detail and introduce related materials and the latest preparation techniques.

### 2.1. Flexibility and Toughness

As an electronic device that mimics human skin, it is fundamental to mimic one of the important properties of skin, flexibility, as it allows the e-skin to bend and stretch freely over a wider range without deformation or damage, thus ensuring free movement of the human body and providing protection to a certain extent. The enhancement of flexibility and stretchability is usually more basic, and the main strategies can be realized through the geometric design of rigid materials and the use of new flexible materials. While it is challenging to prepare materials that are both tough and flexible, like human skin, muscle tissue, and even tendons, it is possible to mimic the flexibility of the human body by designing the cross-linking structure within the flexible material.

One strategy that is often used to achieve stretchability out of rigid materials is to design their geometry. Device structures and electrodes are often designed as, for example, serpentine, petal-shaped, origami, island chain structures, and braided structures, and these different structures ensure that the electrical properties of the device are guaranteed under permissible deformations [19,20,21,22,23,24,25,26,27,28,29,30]. The wavy device structure is usually the most common (Figure 2a), and this structure is achieved by pre-stretching the planar surfaces [31]. Although the stretchability of the devices prepared using this strategy is very limited, the wavy structure is widely used due to its simple operation [21,31,32,33]. The serpentine-structured electrodes, petal-like electrodes, and helical electrodes provide excellent stretchability, and the functional devices are connected through the stretchable electrodes to form a classical island-bridge structure, which is a design that ensures that the functional devices remain relatively independent of each other to stabilize their performance during the bending and deformation of the overall device [34,35,36,37,38,39]. For example, Xin et al. designed a serpentine–honeycomb conformal structure of electrodes that maintains the high ductility of the serpentine electrodes and ensures low impedance, and the interconnect for tissue hemodynamic detection can transmit high-frequency signals and parasitic capacitance (Figure 2b). The serpentine–honeycomb structure follows the deformation law of honeycomb structure at the beginning of deformation, and the serpentine part can relax the stress when the tension reaches the honeycomb limit. The square impedance of the serpentine–honeycomb structure remains at 2.6 Ω at 30% strain (the maximum practical value for skin tissue) (Figure 2c), which fully satisfies the real-time monitoring behavior of the device on the hemodynamics of human tissues when it is attached to the skin and operates normally under various activities in the human body [26]. However, the geometrical design of the rigid material, although it can satisfy the demand for flexibility, its intrinsic non-stretchability and limited flexibility greatly limit their more multifaceted applications. New advanced materials are gradually moving away from traditional rigidity and adopting elastomers with excellent flexibility instead. A large number of stretchable soft materials have been demonstrated for use in e-skin health monitoring devices. The conformal contact of soft materials can produce a high signal-to-noise ratio, unlike rigid materials, dramatically improving the accuracy of the acquired signal. Popular elastomeric materials include polydimethylsiloxane (PDMS), styrene ethylene butylene styrene (SEBS), and polyurethane (PU) [19,40,41,42,43,44,45,46,47], which can have a maximum elongation of more than 1000%. These elastic polymers, together with the corresponding conductive material fillings, can realize a variety of complex microstructures for high-precision sensing. For example, Kong et al. developed a maskless fabrication via selective laser activation to create arbitrary liquid metal features as highly conductive and super-stretchable conductors [46]. The liquid metal possesses both the fluidic properties of a liquid and the high conductivity of a metal, among other properties, which, combined with a highly stretchable substrate, can realize the electrical stability of the device under ultra-high stretch at 1000% strain.42 Gao et al. presented a multilevel structure of helical carbon nanotube/PU yarns with a maximum stretch of up to 1700% (Figure 2d) [41]. This ultrahigh strain capacity is attributed to the helical geometric design and the intrinsic stretchability of the PU elastic polymer. At the same time, the helically structured PU can tightly bind the carbon nanotube lattice and thus exhibit good electrical conductivity and maintain stable electrical properties under 900% strain.

The above demonstrates advanced flexible and stretchable materials, and the current e-skin based on them is fully capable of meeting the bending and stretching caused by the daily movements of the human body. However, the development of materials based on flexibility and stretchability with high toughness to match the Young’s modulus of tissues such as human skin, muscle, and tendon is still a pressing problem. Hydrogels, especially dual network (DN) crosslinked hydrogels, have become the material of choice for e-skin that is simultaneously highly flexible and highly tough due to their excellent stretching row, self-entertainment, biocompatibility, and tunable toughness [48,49,50,51,52,53,54]. The DN structure consists of two polymer networks with opposing mechanical properties, which interpenetrate each other through the introduction of reversible crosslinking, where one of the networks is highly stretched and densely crosslinked to exhibit hard and brittle properties, and the second network is pliable and sparsely crosslinked, making it soft and stretchable. These two different properties fully correspond to the soft and tough nature of human muscle tissue. When subjected to stress, internal fracture of the brittle network dissipates a large amount of energy during large deformations, whereas the flexible network provides elasticity to maintain the integrity of the hydrogel [48]. Employing a freeze-cross-linking strategy is an easy way to obtain DN-structured hydrogels with high flexibility and toughness. PVA ionic gels prepared by the freeze–thaw method can manipulate Young’s modulus so that they are far more flexible than traditional PDMS and SEBS. As shown in Figure 2e, a poly(vinyl alcohol) (PVA) hydrogel framework with a 3D ordered honeycomb structure was obtained due to the directional growth of ice dendrites, followed by aniline cold polymerization along the PVA hydrogel framework to form poly(aniline) [PANI] nanofibrous branched [49]. The Hoffmeister effect proposes that a change of the polymer aggregation state can be achieved by the simple addition of specific ions, where different ions have different polymer precipitation capabilities [55,56]. Also, hydrogels can be made anisotropic in structure at larger scales by different freeze-solutions, and molecular concentration can be facilitated. He et al. used a combination of molecular and structural engineering approaches to fabricate hydrogels whose Young’s modulus could match that of human tendons [57]. By combining directional freeze-casting and subsequent salting-out treatments, hydrogels with varying lengths of scale from the millimeter scale to the molecular level synergistically create hydrogel structures at different length scales. The freeze–thaw crosslinked hydrogels can achieve an ultimate stress of 23.5 ± 2.7 MPa, a strain of 2900 ± 450%, a toughness of 210 ± 13 J/m^3^, an energy of rupture of 170 ± 8 kJ/m^2^, and a fatigue threshold of 10.5 ± 1.3 kJ/m^2^, which comprehensively matches or even outperforms the mechanical properties of human muscle tissue.

### 2.2. Conformability and Breathability

In order to ensure that the e-skin can be worn for long periods of time without interfering with normal human life, conformability and breathability are of paramount importance [18,31,58,59,60,61,62,63,64]. Conformability ensures that the body is free to move without being constrained by the device and that signal acquisition by the device is not affected by movement. Because the e-skin conforms to irregular and complex skin surface shapes, this allows it to fit snugly to human skin or other curved structures, providing a more natural feel without causing discomfort to the wearer, which is important in wearable devices for medical monitoring or health surveillance. Conformality also allows the sensor to more accurately capture subtle changes in the surface, reducing data errors and improving detection sensitivity and accuracy. A conformal e-skin must be highly flexible and able to withstand repeated bending, stretching, or twisting without damage, which is critical for real-time health monitoring in dynamic environments. When the e-skin can be precisely matched to the surface, the external stress concentration points are reduced, thus reducing mechanical fatigue and breakage due to unevenly distributed stress, which can effectively prolong the lifespan of the e-skin, especially in long-term wear or complex motion applications. Usually, the thickness of flexible materials can be reduced by making them conformal; for example, Pan et al. prepared an ultrathin substrate of 1.1 μm by high-temperature cracking using an ultrathin poly(parylene-C) film deposited on a silicon wafer support, and an ultrathin device with conformal properties was formed by stripping the ultrathin substrate from the silicon wafer support using electrochemical layering (Figure 3a) [31]. The overall weight of this ultrathin device was only 3.12 g m^−2^, and attaching the device to the finger enables close contact consistent with the morphology of complex patterns such as fingerprints and does not delaminate from the skin when subjected to bending and compression (Figure 3b).

Breathability is an important indicator of the comfort of wearing an e-skin, and when it comes into contact with biological interfaces, especially in sensitive skin or wound areas, breathability can greatly inhibit the exacerbation of injuries or skin sensitization. Usually, breathability is realized by porous materials, and common porous materials will be prepared using a simple sacrificial template method. Zhang et al. prepared e-skin with 3D spherical shell network carbon nanotubes with porous junctions by using argillic sugar particles as a sacrificial template, and the sensor has −1.2/kPa (pressure resistance) and 0.38/kPa (pressure capacitance). The sensor has high sensitivity and a wide pressure sensing range of 1–520 kPa [65]. More importantly, it is breathable and biocompatible, so that it is worn on the human body like bare skin and is not accompanied by a rise in temperature. However, one of the obvious shortcomings of this method is that it is difficult to prepare thin materials. In recent years, e-skin can be greatly improved by preparing porous fiber materials by electrostatic spinning or by directly using natural fiber-based materials with good biocompatibility. Li et al. prepared a porous stretched thermoplastic polyurethane porous mesh support layer by electrostatic spinning, which was combined with Ag nanowires and thermoplastic polyurethane/Ag nanofibers to construct a patterned nanofiber composite porous ultrathin film (Figure 3c) [66]. The film possesses better air permeability than commercial cotton, and it achieves air permeability with a thickness factor (GF) of up to 88 over a 79% strain range and stable electrical properties at 110% strain. Zhang et al. constructed a unique, ultrathin, ultra-lightweight, and air-permeable array of electrostatically spun microcones by electrostatic spinning self-assembly, which has a mass of only 1.1 mg per square centimeter and consists of loosely packed micro/nanofibers consisting of porous microcone films with excellent breathability [67]. The ultralight, ultrathin device using electrostatically spun nanofiber adhesive showed no change in the appearance of the skin of the wearer’s fingertip after being pressed onto the fingertip for seven hours through humidification treatment, and 95% of the volunteers reported that the device had no impact on their daily lives. In contrast, devices based on conventional PDMS films had to be taped to the fingertips, and after seven hours, the skin of the wearer of the PDMS film device became wrinkled and whitened (Figure 3d). In addition, porous films prepared by self-assembled electrostatic spinning allow optimization of the structure and materials of the device for daytime radiative cooling, pressure sensing, and bioenergy harvesting while wearing on the skin, while guaranteeing comfortable wearability that is ultrathin, non-sensitive, and breathable. The radiatively cooled fabric achieves a temperature drop of ~4 °C at a solar intensity of 1 kW m^−2^, and its high sensitivity (19 kPa^−1^) enables the detection of ultra-weak fingertip pulses for health diagnostics, ultra-low detection limit (0.05 Pa), and ultra-fast response (≤0.8 ms).

**Figure 3 biomedicines-12-02307-f003:**
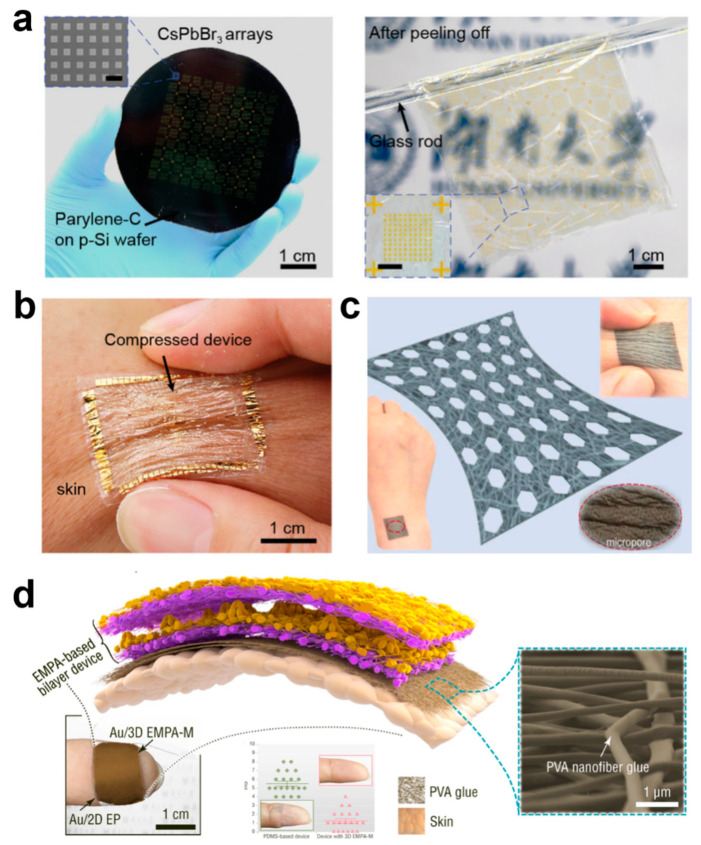
**Conformability and Breathability of Ultra-Thin E-Skin Devices.** (**a**) Preparation and Stripping of Ultra-Thin Devices [1]. Copyright 2021, John Wiley and Sons; (**b**) Ultra-thin device adheres to skin [31]. Copyright 2021, John Wiley and Sons; (**c**) nanofiber composite porous breathable film [66]. Copyright 2023, Elsevier; (**d**) Ultra-light and ultra-thin devices with electrostatically spun nanofiber adhesive [67]. Copyright 2022, Springer Nature.

### 2.3. Degradability

With the widespread use of electronic devices, e-waste places a huge burden on the environment. Metals and plastics in traditional electronic devices are difficult to degrade, and improper handling can lead to pollution, while the disposal of e-waste consumes a lot of manpower, material, and energy. Biodegradable e-skin can naturally decompose after use, effectively reducing the impact of e-waste on the environment and alleviating unnecessary energy consumption, which is more in line with the concept of sustainable development in the new era of technological environment. On the other hand, in the field of health monitoring and medical treatment, e-skin can be used to monitor the patient’s physiological signals or even as an implantable device. Biodegradability ensures that the skin surface or human implant can be naturally degraded in the body after the monitoring task is completed, avoiding the need for a second surgery to remove the device. This non-invasive approach is safer for patients and reduces the cost and risk of medical intervention. Some e-skins are designed for short-term monitoring or tasks, such as post-surgical rehabilitation monitoring, temporary sensors in trauma recovery, etc. The automatic decomposition of the e-skin after the short-term completion of the task avoids the hassle of keeping it in the body or environment for a long period of time. Most importantly, whether naturally degradable or biodegradable e-skin, it usually adopts environmentally friendly, non-toxic, and biocompatible materials to replace heavy metals and harmful chemicals contained in traditional electronic devices, making it safer, especially when it comes to the human body or living organisms, and will not bring about residuals of hazardous substances and health hazards. Chitosan (CS) is the product of removing some of the acetyl groups of the natural polysaccharide chitin, which is the second largest natural polymer after cellulose and is considered to be one of the most valuable natural polymers, so CS, which is antimicrobial, antifungal, antiviral, nontoxic, fully biocompatible, and biodegradable, as well as having film-forming, fiber-forming, and hydrogel-forming properties, is one of the most important materials in the preparation of biodegradable e-skins [68,69,70,71,72]. Xiao et al. proposed a multifunctional biodegradable e-skin utilizing microcracked structured CS membranes as the dielectric layer and substrate and interlaced gold nanowires as the intermediate electrodes [68]. In this e-skin, the CS membranes have excellent optical transparency (94.8%), good flexibility (elongation of 87% and a Young’s modulus of 1.3 MPa), high sweat vapor permeability (water vapor permeability of 1.91 kg m^−2^d^−1^) and degradability, and the gold nanowires ensure stable electrical properties during stretching and bending. Since chitosan is a natural polysaccharide that is easily degraded and decomposed into oligomers during the degradation process, the CS film gradually changed from transparent to yellow in aqueous pepsin solution and achieved complete degradation after 30 h of incubation at 37 °C (Figure 4a). In addition, some synthetic polymers are also biodegradable, such as polyvinyl alcohol (PVA), polylactic acid (PLA), and polylactic acid-hydroxyglycolic acid copolymer (PLGA) [73,74,75,76,77]. For example, Liu et al. used a fully biodegradable e-skin consisting of natural leaf veins, a film of PLGA/PVA nanofibers, and a composite film composed of the natural porous structure of the leaf veins and that of the electrostatically spun nanofibers to provide the sensor with a superior biodegradability. The composite film composed of leaf veins and electrostatically spun nanofibers gives the sensor excellent flexible air permeability (Figure 4b) [76]. The PVA fiber film undergoes rapid autocatalytic hydrolysis and volumetric degradation when exposed to water, with a weight loss rate of almost 100%.The PLGA fiber film initially has greater resistance to weight loss and water absorption, but due to the hydropyrolysis of the polymer backbone, it shrinks and curls up slightly, and after about 20 days, the PLGA fiber film degrades to 50%. The degradation of the leaf veins showed a relatively slow rate throughout the process, and a faster degradation could be achieved by the addition of a complex cellulolytic enzyme, and finally the device degradation was completed in 45 days outside the room. Peng et al. prepared a multistage, multilayer, multihollow-structured e-skin by intercalating silver nanowires between PLGA and PVA [77]. By adjusting the concentration of the silver nanowires as well as the z-component of PVA and PLGA, the antimicrobial and biodegradable capabilities of the e-skin could be adjusted, respectively. In conclusion, the novel materials with flexibility, breathability, and biodegradability provide comfortable, safe, and non-polluting e-skins with health monitoring and medical functions, which can help to promote safer, longer-lasting, and more environmentally friendly applications of e-skins in human–computer interfaces and artificial intelligence.

## 3. Fundamentals and Technical Components of E-Skin for Healthcare

### 3.1. Sensing Mechanisms and Functions of the E-Skin

The different sensing mechanisms of e-skins determine their different performance for health monitoring and medical fields. Currently, e-skins not only have the sensing functions of stress–strain, body temperature, humidity, gas, biochemical, and biophysical signals but also have many other functions, such as energy harvesting, self-healing, refrigeration, light-heat conversion, and drug slow-release. The different functions of e-skins have greatly enriched and expanded their applications in the healthcare field and enhanced their practicality. Mechanical force sensing functions such as stress–strain are the most important, extensive, and basic function of e-skin. According to the sensing mechanism, they can be categorized into five types: resistive, capacitive, piezoelectric, and triboelectric (Figure 5a) [2,78,79,80,81,82,83,84,85,86,87,88,89]. Resistive e-skin is sensed by the change in resistance value caused by deformation of the material under the action of an external mechanical force [2]. Under the same external force, the change in contact resistance is much larger than the intrinsic resistance, resulting in an increase in sensitivity. For most contact-resistive e-skins, rising pressure decreases the resistance, resulting in a negative resistance change [81]. Resistive e-skins are typically composed of intrinsically conductive elastomeric, stretchable materials or composites of conductive materials with elastomeric, stretchable materials [81,82]. Intrinsically conductive elastomeric, stretchable materials commonly used to prepare resistive e-skins include conductive polymers such as ionic hydrogels. gels and other conductive polymers. Common polymer substrates for resistive e-skins include PDMS, Ecoflex, PVA, and styrene–ethylene–butylene–styrene block copolymers. Conductive substrates commonly used for compositing with polymers include liquid metals, metal micro/nanoparticles, metal films, metal nanowires, graphene, carbon black, carbon nanotubes, and so on. For example, Zhang et al. proposed a hydrogel-based piezoresistive sensor whose strain-resistant effect is attributed to the change in free ion conduction in the three-dimensional porous structure during deformation [90]. When the sensor is stretched, the cross-sectional area decreases, which blocks the conductive pathway for the free ions, leading to an increase in resistance. Resistive electronic skin is simple in structure, easy to prepare, has high sensitivity and low hysteresis, and is one of the most commonly used flexible sensor devices today [82,91,92,93]. It usually consists of two parallel electrode plates. When an external force is applied to the sensor, the geometrical parameters of the capacitive sensor, such as the distance between the electrodes and the relative area of the dielectric layer, change, which leads to a change in capacitance [94]. Conducting materials such as metal thin films, metal nanowires, graphene, carbon black, carbon nanotubes, etc. have been widely used to prepare the electrode materials of capacitive e-skins, whose dielectric layer is usually composed of elastomeric materials with a micro/nanostructure. dielectric materials with micro/nanostructures. Obviously, capacitive e-skins have a more stable baseline compared to resistive ones, are easy to achieve higher linearity, and have high sensing reliability; however, their capacitance values are highly susceptible to interference by environmental parasitic noise [83]. Traditional capacitive and resistive e-skins based on capacitive and resistive responses have been developed for many years with stable performances and a wide range of applications.

Piezoelectric and triboelectric e-skins are relatively new sensing methods. Unlike traditional capacitive and resistive e-skins, these two types of sensing methods have extremely fast response speeds, high output power, and have significant advantages in self-driving. Piezoelectric e-skins are sensed by external mechanical forces that cause a change in the dipole deflection or dipole moment of the material, resulting in a change in the piezoelectric potential on the surface of the material [95]. Typically, non-centrosymmetric crystals are piezoelectric in nature. Of the 32 crystallographic point groups, 21 have non-centrosymmetric structures, of which the crystals in point group 432 are not piezoelectric due to other symmetry factors, and the remaining 20 have piezoelectricity [84]. Conventional piezoelectric materials include crystals with the structure of fibronzincite, such as zinc oxide and gallium nitride, and ferroelectric crystals with the structure of chalcocrystalline titania, such as barium titanate, lead zirconate titanate, and sodium and potassium niobates [85,86,96,97,98,99]. Compared with resistive electronic materials, these materials are more efficient in terms of the piezoelectricity they can produce than resistive electronic materials. Compared with resistive and capacitive electronic skins, piezoelectric electronic skins have the advantage of ultra-fast response speeds, with response times of no more than tens of milliseconds. With the development of flexible electronic devices, flexible piezoelectric polymers such as polyvinylidene fluoride (PVDF) and its copolymers poly(vinylidene fluoride-trifluoroethylene) (P(VDF-TrFE)) and poly(vinylidene fluoride-hexafluoropropylene) (P(VDF-HFP)) have been widely developed for applications [87,100,101,102,103]. Piezoelectric polymers have a high degree of flexibility, with a flexibility coefficient tens of times that of piezoelectric ceramics. The piezoelectric polymer materials have high flexibility, with a flexibility coefficient that is tens of times that of piezoelectric ceramic materials, and thus can be made into large and thin membranes and have high mechanical strength and toughness to withstand large impact forces. Triboelectric e-skins are based on the coupling effect of contact initiation and electrostatic induction and utilize external mechanical force to cause electrode potential difference for sensing [104,105,106]. Triboelectric e-skins usually have four modes of operation, which are vertical contact-detachment mode, horizontal sliding mode, single electrode mode, and independent-layer mode (Figure 5b) [88]. The vast majority of materials, regardless of whether they are solid, liquid, or gaseous, can be used as sensing devices, and the majority of materials, regardless of whether they are solid, liquid, or gaseous, can be used as electrostatic sensing devices. Most materials, whether solid, liquid, or gaseous, produce a frictional electric effect when they come into contact with each other, so triboelectric e-skins offer excellent material versatility. Triboelectric e-skins have a higher power output than piezoelectric e-skins. However, triboelectric e-skins are very sensitive to humidity changes in the external environment, which can significantly affect their electrical output performance. Since triboelectric e-skins need to go through a charge accumulation process to work properly, the accumulated charge is very easy to dissipate when they are not in operation, and the stability of force sensing in their practical environments is weaker [107]. The triboelectric e-skins have a higher output power than the piezoelectric e-skins, and they are more sensitive to humidity changes in the external environment.

Although excellent mechanical sensors can monitor the daily state of human behavior and emotions and human life indicators such as blood pressure, pulse, respiration, etc., which can basically satisfy the e-skins’ monitoring of human health conditions. However, more precise and specialized physiological indicators and medical parameters cannot be obtained entirely through mechanical sensors. Therefore, in addition to the mechanical force sensing function, e-skins also need to sense temperature, humidity, gas, biochemical molecules, and light. For example, body temperature is a very important physiological indicator. For temperature-sensing e-skins, this type of electronic skin can provide early warning to avoid abnormal temperatures caused by health and safety risks. The dominant mechanism of operation is the resistive temperature sensor e-skins [108,109]. This is due to their ease of flexibility, simple preparation process, compatibility with a variety of substrates or materials, and ease of attachment to the surface of the skin and a variety of materials. Ionic hydrogels and ionic gels are popular materials for the fabrication of electronic skins for resistive temperature sensing due to their excellent stretchability, flexibility, surface fit, and biocompatibility [110]. A wearable ionic gel resistive temperature sensor based on ionically crosslinked polyacrylamide-sodium alginate hydrogel was prepared by Pan et al. The sensor has a response time of only 2.02 s from ≈21 °C to 60 °C, with a sufficiently high temperature difference accuracy of 0.9 °C. Using an array of this temperature sensor applied to the surface of the skin allows for the detection of the temperature distribution of the internal blood vessels at the wrist, enabling fast, high-precision, and high-resolution temperature recognition (Figure 6) [111]. For humidity, gases, biochemical molecules, and light-sensing e-skins, common humidity-sensing e-skins can be similarly categorized into resistive, capacitive, and triboelectric humidity sensors as mechanical sensors [112,113,114]. E-skins prepared with hydrophilic materials, such as polyvinyl alcohol and hydroxyethyl cellulose, are often used for better humidity sensing performance. Common gas-sensing e-skins are resistive and triboelectric gas sensors [115,116]. Biochemical molecule-sensing e-skins are usually resistive [117]. The core component of optical sensing e-skins is the photosensitive element with photovoltaic effect [118,119]. In short, e-skins with various sensing functions are used in human-computer interaction, robotics, biomedicine, safety and security, etc. E-skins with various sensing functions provide new possibilities and solutions.

### 3.2. Indicator Parameters of E-Skins in Pan-Health Management

E-skins used in areas such as health monitoring and medical treatment need to have clear indicators of the tested physiological parameters to analyze and predict the health of the human body. The most important health indicators of the human body, which also evaluate the presence or absence of life activities and their quality, are the vital signs: temperature, respiration, pulse, and blood pressure. E-skins need to continuously monitor these signals in daily life and send out the interactions of the different signals to differentiate between the signals that need to be monitored in order to provide preliminary diagnostic information.

#### 3.2.1. Body Temperature

In the monitoring of vital signs, body temperature is an important parameter for detecting physical abnormalities and providing medical diagnosis [120,121]. The body’s oral (36.5–37.5 °C), axillary (36–37 °C), and rectal (37–38.1 °C) temperatures need to be maintained at specific temperatures to maintain stable body functions. However, the usual use of thermometers to measure these areas of the body is uncomfortable and does not allow for real-time monitoring. Furthermore, temperature variations occur in individuals of different ages, at different times of day, during exercise, in the external environment, and in different hormonal states. Therefore, a flexible temperature sensor can be attached to the skin surface for a long period of time to achieve continuous temperature monitoring, and real-time analysis and prediction in conjunction with the actual situation can help to accurately determine the health status of the human body [122,123,124,125]. Since it needs to be worn for a long period of time for real-time monitoring, the sensor requires a high degree of flexibility to fit to the skin and to withstand deformations, and it needs to be highly accurate in monitoring the changes in the body temperature and at the same time insensitive to the external ambient temperature. Bao’s team developed a temperature sensor with a sensitivity of up to 0.3 V/°C and a maximum error of ±3.1 °C (3σ), and in particular, the sensor has a strong positive temperature coefficient effect between 35 °C and 42 °C, which is well adapted to human skin temperature [126]. Ryu et al. developed a flexible temperature sensor consisting of three layers of multimaterial rod-like polymer nanocomposite fibers. composite fibers [127]. The innermost portion of the temperature sensor fiber is a temperature sensing core composed of temperature-dependent resistive reduced graphene oxide (rGO) and the thermoplastic biocompatible polymer polylactic acid, and the outer layer consists of linear low-density polyethylene (Figure 7). The sensor exhibited a sensitivity of −0.285%/°C at 25–45 °C with fast response and recovery times (11.6 s and 14.8 s). Since the temperature sensor is composed of composite fibers, the adjustable diameter allows it to be woven into fabrics or sewn into garments such as shirts and gloves for real-time body temperature monitoring.

#### 3.2.2. Respiration

Among human vital signs, respiration is a key indicator of the body’s respiratory system and overall health. Health is usually analyzed by monitoring respiration from the following aspects: respiratory rate, respiratory depth, respiratory pattern, and oxygen saturation (Figure 8a) [128]. The respiratory rate in normal adults ranges from about 12–20 breaths/minute and is slightly higher in children. Abnormal respiratory rate can indicate underlying health problems, e.g., too fast respiratory rate (shortness of breath) may be associated with anxiety, hypoxia, fever, acidosis, etc., and too slow respiratory rate may be associated with medication overdose, neurological disorders, etc. Respiratory depth is the amount of air inhaled or exhaled with each breath, which is normally about 500 mL, and its variation may reflect airway obstruction, lung disease, or other disorders affecting respiratory muscle function. Breathing pattern refers to the rhythmic and even nature of breathing. Normal breathing is usually smooth and silent. Abnormal breathing patterns (e.g., tidal breathing, Chen–Schiff breathing, etc.) may indicate serious neurologic or cardiopulmonary dysfunction. Oxygen saturation is the amount of oxygen carried in the blood and is usually measured by a medical pulse oximeter; normal oxygen saturation should be between 95% and 100%. Breathing is an essential life-sustaining function, and monitoring respiratory indicators allows for early identification of potential health problems, especially acute and chronic respiratory, cardiovascular, and metabolic diseases. By detecting parameters such as respiratory rate and oxygen saturation, deterioration can be detected in time, helping healthcare professionals to take rapid and effective interventions to improve patients’ survival and quality of life. However, usually respiratory monitoring cannot do real-time and multi-faceted synchronous measurement; only wearing medical respiratory equipment can real-time monitoring, which seriously affects people’s normal behavioral activities. Therefore, the development of flexible respiratory monitoring e-skins is of great significance in the healthcare field [128,129]. Sasaki’s team fabricated a wearable e-skin for monitoring human respiratory rate, which is used to measure respiratory flow, frequency, and recognize breathing patterns. Capacitance measurements between two electrodes placed on the inside front and back of the t-shirt in the chest position allowed monitoring of the respiratory cycle [130]. Ni et al. designed a highly sensitive humidity hydrogel sensor for respiratory monitoring; the sensor exhibited a fast response over a humidity range of 40–85% RH and had a −103.4%/% RH with high sensitivity [131]. By monitoring humidity changes in the respiratory airflow, a corresponding respiratory cycle signal response curve was derived. The sensor can monitor breathing during exercise and sleep, and the respiratory sensor can detect respiratory changes in the mouth and nose, pressure changes in the chest/abdomen, and apnea during sleep, a unique idea that provides a new research direction for the application of hydrogel-based humidity sensors in respiratory monitoring (Figure 8b).

#### 3.2.3. Pulse and Blood Pressure

With the COVID-19 pandemic and the elevated stress of life for humans as a whole, this has led to a significant increase in mortality from chronic diseases, particularly cardiovascular disease, which has become the number one disease causing mortality in humans [132]. Cardiovascular diseases cause more than 17 million deaths globally each year, which accounts for approximately 31% of the total global mortality rate [133]. Although the suddenness and unpredictability of the onset of most cardiovascular diseases Although the suddenness and unpredictability of most CVD episodes cause them to pose a significant risk to humans, 90% of these cases can be prevented through early detection [134,135]. Pulse and blood pressure (BP) are important indicators of the body’s cardiac function, which can be used to assess the functioning of the cardiovascular system and overall health and thus to anticipate and avoid the onset of cardiovascular disease [136]. The pulse and BP are important indicators of the body’s cardiac function. Monitoring pulse and BP reflects the state of the heart and blood circulation and plays a very important role in the prevention and monitoring of cardiovascular diseases [136,137,138]. Traditional home-based heart rate and BP monitoring relies on cuffed sphygmomanometers, and it is clear that they cannot be comfortably worn and allow for long-term sensation-free monitoring. The development of wearable and cuffless pulse and blood pressure monitoring e-skins to achieve long-term uninterrupted and comfortable health monitoring can be effective in avoiding cardiovascular diseases (Figure 9a) [139,140,141]. For example, Kim et al. developed a soft capacitive pulse and blood pressure monitoring e-skin based on a highly wrinkled Au film [142]. Using a micro-ridge structure supporting a pair of electrodes, an air cavity is formed within the elastic dielectric layer, and the pressure sensitivity can be as high as 0.148 kPa^−1^ over a wide dynamic range as high as 10 kPa. The fast response time of less than 10 ms Pan’s team prepared a pulse monitoring sensor based on a perovskite thin film array through a selective vapor deposition strategy [143]. The excellent optoelectronic performance (13,877 on/off current ratio, 0.81/2.03 ms response time) and long-term durability (12 h of high humidity) give the sensor the ability to continuously monitor the pulse rate. The ability of the sensor to continuously monitor the pulse. Conformal contact of the device with the finger reduces the effect of optical noise and allows high-precision detection of the systolic and diastolic peaks of a single pulse, and multi-pixel detection improves the quality of the pulse signal by effectively eliminating signals generated by jitter or slippage during the test (Figure 9b).

## 4. Energy Harvesting and Data Transmission

### 4.1. Energy Harvesting Technology

Energy harvesting technology is the technology that ensures that e-skins can be worn for long periods of time without sensation and for continuous health monitoring. The energy harvester introduced can be used as a power source for the sensors or directly as a self-powered sensor, eliminating the need for an external power source [18,144]. Energy harvesting technology used as e-skins is usually self-powered, which effectively avoids the need for external power sources for traditional electronic devices. to realize that e-skins can be worn for long periods of time without affecting normal life. The energy used in e-skins is divided into many types, including mechanical, photovoltaic, thermoelectric, and chemical energy, of which mechanical energy is the most common and easiest to harvest. Currently, friction nanogenerators (TENGs) and piezoelectric nanogenerators (PENGs) are the dominant self-powered systems for harvesting mechanical energy to provide energy for e-skins. TENGs are an innovative energy storage technology that efficiently utilizes a wide range of ambient energy sources as well as energy dissipated by human movement. This includes converting mechanical energy generated by human behavior (movement, breathing, vocalization, etc.) and various energy sources generated by the external environment (rain, wind, waves, etc.) into viable power sources, making them the subject of extensive research for applications such as wearable power sources and self-powered sensors [145,146,147,148]. Since its introduction by Prof. Zhonglin Wang in 2012, TENG technology has made significant progress in the understanding of the underlying mechanisms and the development of self-powered systems. and the development of self-powered systems has made many significant advances [149,150,151]. The four basic modes of TENGs were introduced in the previous section, and TENG-based self-powered systems are also derived from these four modes, with most of them realizing self-powering of the system through the vertical contact separation mode [152]. Recently, Fan et al. developed a novel TENG technology without a conductive layer as a self-powered sensor [153]. This sensor can not only collect energy through the vertical contact mode but can also efficiently collect energy when the two triboelectric layers are bonded together and cannot be separated by contact or sliding. When the two layers of TENG are deformed (stretched, folded, pressed with a pen, etc.), the charge is redistributed and transferred. Due to the difference in frictional electrical properties, the electronic balance is upset, and electrons move from one layer to the other and generate an electrical signal that is collected and output via a silver wire attached to one layer. This is also referred to as the fifth mode of energy harvesting by TENGs that differs from the traditional four modes—the deformation mode (Figure 10). Zhang et al. proposed a core-shell porous fiber TENGs based on liquid metal and PDMS, which can be directly compiled into a textile to provide sustained energy for the e-skin [154]. Using an in situ foaming strategy to achieve high porosity and improve the friction electrical properties on the surface of fibrous TENGs. Fine tuning of the fiber shell thickness and surface aperture can amplify the output V_OC_ by a factor of five and the output I_SC_ by a factor of seven. A brooch-sized fiber TENG generates an alternating current that can charge a 10 μF capacitor to ~7 V at 4 Hz in 250 s. Huh. et al. innovatively developed biocompatible TENG made of biocompatible chitosan films for the fabrication of edible electronic skin. The TENG device has a maximum electrical output of 295.3 V, 4.1 μA, and can power commercial capacitor charging and low-power electronics such as LEDs [155].

Similar to the energy harvesting strategy of the TENGs, the PENGs can also directly convert mechanical energy into electrical energy to realize a self-powered strategy [156]. The principle of electrical energy harvesting in piezoelectric materials is similar to the sensing principle, which has been elaborated in the previous section. When external pressure is applied to the PENG-based e-skins, it causes the piezoelectric material to deform, resulting in a negative strain and a relative decrease in volume. Separation of the charge centers creates electric dipoles, leading to a change in the electric dipole moment, which creates a piezoelectric potential at the electrodes. Connecting the electrodes to an external load allows the piezoelectric potential to drive electrons through an external circuit, partially neutralizing the potential and reaching a new equilibrium state. After the external force is withdrawn, the electrons return to restore the charge equilibrium induced by strain release under short-circuit conditions [135,157]. Xia et al. recently proposed a strategy to enhance the output performance of PENGs based on core-shell heterostructured fibers by improving the polarization and stress transfer mechanisms [158]. PENGs were prepared by coaxial electrostatic spinning through the mixing of barium titanate@Ag heterostructured particles and PVDF in the barium titanate@Ag in the outer layer of the composite fibers, which enhances the polarized electric field and stress transfer of the piezoelectric filler, and the improved PENG output is increased by about three times. Mahanty et al. proposed a new combination of heterogeneous layer-structured alternately stacked barium titanate nanoparticles of P (VDF-TrFE) and graphene nanosheets-embedded P (VDF-TrFE) piezoelectric composite nanofibers with spacer metal flake design. These parameters cumulatively enhance the Maxwell displacement current, which greatly enhances the main driver of the PENGs collection capability [159]. The self-driven system can achieve an open-circuit voltage output of 350 V, a short-circuit current output of 6 μA, and a power output of 3.62 W m^−2^. The wearable, self-powered, unpowered health-monitoring system provides a wide range of information on human movement, mood, pulse, and other aspects of health. The wearable self-powered health monitoring system can provide remote health monitoring of human movement, mood, pulse, and many other aspects (Figure 11).

In addition, bioenergy from the human body, light energy from nature, magnetic energy, thermal energy, and other energy sources can also become the e-skins energy collection system, but at present, due to the limitations of technology and application conditions, it is far less widely used than the first two self-powered. Bioenergy is considered to be the most promising green energy source. Sweat produced by the human epidermis contains a variety of metabolites, such as lactic acid, glucose, salts, etc., which can be captured by e-skins to feed the biofuel cell when performing the monitoring function [160,161]. Usually, self-powered systems that collect bioenergy use biochemical reactions in living organisms to convert chemical energy stored in organic matter into electrical energy. Electrical energy. Within the system, oxidative enzymes catalyze the oxidation of a biofuel (e.g., glucose or ethanol), producing intermediates that facilitate the transfer of electrons and protons, while oxygen reductase catalyzes the reduction in oxygen, which absorbs electrons and protons from the anode and subsequently combines with oxygen to form water. These electrons produced in this process are released to the electrode surface and flow through an external circuit to power the e-skins or store them for later use (Figure 12) [162]. Veenuttranon et al. demonstrated a single enzyme-based energy harvesting device with the integration of a glucose-driven, self-powered biosensor [163]. The system converts the chemical energy of glucose in sweat into electrical energy, achieving an open-circuit voltage of 0.45 V and an energy density of 266 μW cm^−2^, and can monitor glucose concentrations of 10 mM. Self-powered sensing systems based on the thermoelectric or pyroelectric effect of heat energy harvesting utilize the body temperature gradient or temperature fluctuation for energy conversion and are mainly used for the monitoring of respiration, body temperature, and heart rate during human movement. However, wearable thermoelectric sensors are rarely used for practical applications due to their low power conversion efficiency. Self-powered sensing systems based on the photovoltaic effect of solar energy harvesting have high energy conversion efficiency and also provide effective ideas for energy autonomous systems. Examples include halogenated perovskite materials that are low-cost, simple to prepare, have adjustable bandgaps, and have high photovoltaic conversion efficiencies [164,165,166,167,168,169,170,171,172,173,174]. Current perovskite solar cells (PSCs) can have conversion efficiencies as high as 25.7% [175]. However, due to the conditions of perovskite’s susceptibility to decomposition in high humidity environments, the presence of toxic lead elements, and the inability to prepare flexible devices, etc. limitations, devices using perovskite as an energy harvesting platform in e-skins are not common.

### 4.2. Wireless Data Transmission

Realizing long-term, stable, and real-time monitoring without affecting the wearer’s life has high requirements on the communication performance of e-skins. The ultimate goal of wireless communication technology is to wirelessly transmit dynamic physiological signals to a smart device in a fast and timely manner for professional clinicians to make a diagnosis. Usually, e-skins in the early stage of laboratory preparation are connected to the system through complex and bulky test equipment to collect a series of signals. However, in practical applications, it is not possible to carry as elegant equipment as in laboratories, so wireless transmission systems are needed for real-time, fast transmission of the collected data. Depending on the distance between the sensing device and the data receiver, the way of power supply, functional characteristics, etc., wireless communication technologies such as Radio Frequency Identification (RFID), Near Field Communication (NFC), and Bluetooth Low Energy (BLE) have been developed to be applied to wearable devices, which provide ideas for real-time monitoring of the device, dynamic response, and instant feedback [176,177,178,179,180,181,182,183]. RFID is a two-way communication technology with a unique identifier. for two-way communication and can be equipped with various sensing functions. Due to these properties, RFID plays a key role in people’s daily lives and industries, such as cargo tracking, food safety, environmental sensing, security, and many other areas [184,185]. Bao’s team reported a body-area sensor network consisting of chip-less and battery-less stretchable sensor tags using unconventional detuned RFID tag design to handle the antenna inductance and resistance’s strain-induced changes [186]. The e-skin flexible sensor tags have no chips or batteries inside, and the network concept is realized by multiple sets of flexible sensors and soft readout circuits located on the clothing. The monitored signals are transmitted via RFID to a cell phone, enabling real-time monitoring of human biophysical signals (breathing, pulse, etc.).

NFC is based on RFID technology combined with wireless interconnect technology and is a subset of RFID. NFC uses only the 13.56 MHz band of RFID’s different frequency bands to realize fast communication of devices over short distances [187,188]. The antenna is integrated with e-skins for battery-free passive mode and contactless data transmission and wireless communication. Compared to RFID, NFC technology offers high security and confidentiality and is compatible with e-skins for battery-free passive mode, contactless data transmission, and wireless communication [187,188]. The antenna is an important part of the e-skins that integrates NFC, and its main function is to transmit the RF signals between the tag and the reader in order to realize real-time wireless transmission of data. Due to the need for long-term wearable requirements, therefore, NFC tags must not compromise the comfort of the wearer under any circumstances, which requires not only a flexible, compact, and lightweight design for reasonable integration but also sustainable operation to avoid frequent recharging. In addition, a stable and controllable wireless communication link is required to provide the wearer with freedom of movement without going out of reading range [189]. Bandodkar et al. reported a wearable sweat sensing platform that consists of a flexible microfluidic network and a lightweight wearable NFC electronic module [190]. The disposable microfluidic network houses chemical sensors to process and analyze small amounts of sweat delivered to the platform through the action of glands, and the NFC electronics module is mounted on a disposable microfluidic system with a releasable electromechanical interface to ensure that it can be reused (Figure 13). Bluetooth technology typically requires external battery power, so even with a miniaturized signaling system, it is inevitable that the continuous monitoring and comfortable wearing of the e-skins will be compromised. In conclusion, as part of e-skins for health monitoring applications, wireless communication technology is required to seamlessly transmit important information to the user. Ultra High Frequency RFID (UHF RFID) has a long transmission range, but readers are very expensive (1000 USD–2000 USD) and are more susceptible to environmental influences that can cause loss and detuning. Low-frequency RFID (LF RFID) has excellent anti-jamming properties, has been commercialized on a wide scale and is growing rapidly, and has a wide range of applications in the marketplace, but its drawbacks are relatively low transmission speeds and a short operating distance (50 cm). BlE (3–200 m) is the current best choice to ensure real-time and fast communication of large amounts of data. BLE wireless communication has a larger read range and requires less power consumption than NFC; however, large volume and incompatible flexible substrates are still the existing problems. Table 1 summarizes several wireless communication technologies, but the choice of wireless communication technology for e-skins needs to be tailored to the function of the sensor, the specifics of the monitoring, and the population to which it will be applied.

## 5. E-Skin in Pan-Health Applications

### 5.1. Health Management and Exercise Monitoring

The biggest advantage of using e-skin for human health management is that e-skin is able to continuously acquire human health and exercise data in a non-invasive way and provide instant feedback, which is of great significance in disease prevention and exercise optimization. Traditional medical devices are unable to monitor the physiological and exercise data of different individuals over a long period of time. E-skin enables personalized health management programs and helps athletes or ordinary people to develop scientific exercise plans. E-skin is equivalent to having a personal health management assistant to help people monitor and customize personalized private management programs. Breathing, temperature, pulse, and blood pressure–the four important vital signs mentioned in the previous chapter –are very important for everyone’s health management. In addition to this, analyzing sweat is also an important parameter for personal health management and exercise testing, as it can provide metabolic information to help assess fluid balance, blood sugar levels, exercise fatigue, etc. by testing the composition of sweat. Analysis of sweat can also be used to monitor early signs of dehydration, electrolyte imbalance, or other metabolic disorders. For example, Yin et al. self-regulated a biofuel self-powered e-skin sweat sensing system [191]. The system integrates three types of potentiometric sensors, including pH sensors, AA sensors, and LA sensors, to measure representative biomarkers reflecting the user’s electrolyte levels, nutritional levels, and fatigue levels, respectively, during exercise, which can help to provide a comprehensive understanding of the user’s real-time health status and prevent excessive exercise (Figure 14a). E-skins can also perform gait analysis, fatigue detection, joint angle analysis, sitting and standing posture analysis, and sleep analysis, thus assessing human health in a comprehensive way and improving the quality of work and life [192]. For example, Zhao et al. developed a self-powered gait analysis system based on the electrostatically spun composite nanofiber TENG for human movement monitoring and gait analysis, helping users to accurately assess their health status. analysis to help users accurately assess their exercise status [193]. By integrating two sensing units, TENG-S1 and TENG-S2, and an IoT platform, a self-powered real-time pedometer speed gait analysis system consisting of signal processing and wireless transmission was implemented to capture and analyze a large amount of information on the number of steps, touchdown time, walking speed, and acceleration during exercise training and then assess the training on the athletes (Figure 14b).

### 5.2. Emotion Recognition and Mental Health Monitoring

With the growth of the global population and the furniture of the aging population, people are under increasing pressure to focus on the physical health of the individual and at the same time need to simultaneously enhance attention to the heart health, especially in the management of emotions [194,195,196]. Emotions are the human response to events or situations that have a significant impact on our daily lives. However, emotion is a complex psychological state that is not easy to detect and is good at camouflage. Current methods for recognizing emotions include speech signal analysis, image-based facial action coding systems, EEG signal, and electrophysiological signal analysis [197,198,199,200]. Du et al. reported a graphene-mediated, dual-network conductive polymer-based film, with which the facial skin was covered to achieve monitoring of fine EMG [196]. Through precise data acquisition followed by machine learning, the potency was effectively recognized and analyzed (positive or negative) and the arousal (intensity of the emotion) of the emotional experience. Facial EMG signals corresponding to these subtle activities of the facial muscle groups and zygomatic bones are collected through the e-skin. The collected signals were preprocessed using a moving average filter, and then the signals were used to create three databases of 71, 46, and 73 different emotions from a total of 190 facial EMG segments. Subsequently, after calculating its 26 eigenvalues from each segment containing different time and frequency domain information, a total of 190 × 26 eigenvalues can be constructed from all the datasets and used for machine learning, and finally the final classification results are obtained by constructing a classification model using a bi-directional long and short-term memory network (Figure 15). However, emotion recognition and mental health monitoring is a relatively new area of research, including how to implement natural human–computer interaction, how to extract feature values, how to make computer systems that can accurately process and understand human-expressed feelings, and how to efficiently determine changes in emotions [201,202,203]. Emotion computation requires cross-disciplinary knowledge and requires a concerted push from a variety of disciplines to accomplish it, including computer science, artificial intelligence, electrical engineering, psychology, neuroscience, medicine, etc.

### 5.3. Supervision and Care of Infants and Young Children

E-skins have great potential in infant care, providing non-invasive, real-time health monitoring for newborns and infants, helping parents and caregivers to better understand their baby’s health [204]. Infant health monitoring and care need to ensure that the comfort of non-sensory at the same time reduces external interference on the baby. In addition, because babies are smaller and have softer skin, e-skin needs to be small, thin, and biocompatible. E-skins revolutionize infant health management and care, softly conforming to the baby’s skin and providing continuous monitoring of key physiological indicators in a gentle, non-invasive way that does not cause discomfort. The e-skins are soft against the baby’s skin and provide continuous monitoring of key physiological indicators in a gentle, non-invasive way that does not cause discomfort to the infant. In infant care, electronic skin not only improves monitoring efficiency but also reduces unnecessary interventions, providing parents and healthcare professionals with scientific evidence and health assessments and predictions [205,206]. Rogers’ team has developed wireless, battery-free vital sign monitoring systems based on ultrathin “skin-like” measurement modules [207]. These devices can be gently and easily attached to the infant’s skin and provide continuous monitoring of key physiological indicators without causing discomfort. These devices can be gently and noninvasively attached to the skin of newborns of low to marginal gestational age. It consists of two parts that are placed on the infant’s chest and the soles of the feet: the former is used to monitor electrocardiogram signals, while the latter is used to monitor blood oxygen levels, and temperature and respiratory rate monitoring are integrated into the system for continuous monitoring of the neonate’s vital signs. Data are transmitted wirelessly via NFC and BLE to a collection and computer, allowing doctors to remotely monitor the neonate’s condition (Figure 16). Guo et al. achieved accurate and rapid measurement of applied mechanical pressure and recognition of multiple movement patterns of the infant by combining sensors for infant care with the assistance of artificial intelligence and deep learning algorithms [204]. This customized APP program personalizes the monitoring content and data analysis, enabling real-time warnings and one-click guardian interaction. In the future, such infant-oriented smart systems offer a promising and practical paradigm for infant care and management in the IoT era.

### 5.4. Elderly Care

As the global population ages, health monitoring and healthcare for the elderly is a pressing concern [208]. E-skins for elderly care need to focus on monitoring and early warning of prevalent and chronic conditions, sleep quality, post-operative recovery, and assisted independence [89,209,210,211,212]. E-skins automatically capture health data and transmit it via wireless technology, reducing the need for frequent manual testing by caregivers, thus improving the efficiency of care and reducing the burden of care. older adults with frequent manual testing, thereby increasing the efficiency of caregiving and reducing the burden of caregiving. E-skins can also help seniors live independently in their homes, monitor their physical functioning, and provide alerts or help signals to prevent danger and improve the ability of isolated seniors to live on their own. E-skins’ primary goal is to improve the quality of life for seniors by providing non-sensory, continuous health monitoring, assistive behavioral monitoring, and early warning of diseases. For example, custom-made e-skin needs to focus more on behavioral monitoring (falls, wobbles, postures) and real-time alerts (calls for help, alerts, and so on) for older people with reduced mobility. For the elderly with hypertension or obesity, the function of e-skin needs to focus on cardiovascular disease warning. For patients with Alzheimer’s disease, e-skin is required to be able to locate and do environmental monitoring in addition to routine behavioral monitoring. Pan’s team developed ultra-thin, non-sensory, hybrid arteriovenous flow monitoring e-skins for the analysis of clinical symptoms caused by PAD, such as lower extremity ischemia, surgical vascular closure, and skin ulcers [39]. The devices are capable of simultaneously measuring several important parameters, such as rSO_2_, heart rate (HR), arterial oxygenation (SpO_2_), and tissue perfusion (PI) analysis and have demonstrated the ability to correct the ankle–brachial index (ABI), which is the most important parameter for monitoring PAD in older adults. demonstrated the ability to correct false-positive results for ankle–brachial index. Yang et al. connected to a self-powered human–computer interaction system for elderly fingers [213]. The e-skin is attached to the elderly’s finger, which bends as the finger flexes, generating frictional electrical signals and sending them to the caregiver for help, and realizing real-time monitoring and intelligent management in a smart aging system.

### 5.5. Assistive Devices for the Disabled

There is another group of people in this world that cannot be ignored: people with disabilities. Globally, there are nearly 1 billion people with speech, hearing, and visual disabilities who are at high risk for healthcare and life security [214,215,216,217,218,219]. The e-skin should play a unique role in helping people with disabilities by providing assistive functions and health security based on their different physical impairments, family history, lifestyle, and genetics that exhibit unique health conditions. For example, Zhou et al. developed a wearable sign language interpreter system to help deaf people, which can translate sign language into audio in real time [220]. Combining artificial intelligence and machine learning algorithms, the system acquired a total of 660 American Sign Language (ASL)-based sign language gestures and successfully analyzed and interpreted them, and it can achieve a high recognition rate of 98.63% and a short recognition time of less than 1 s, which greatly Pan’s team developed the Integrated Intelligent Tactile System (IITS) based on a multichannel tactile sensing electronic skin, a data acquisition and information processing chip, and a feedback control [221]. The IITS is a closed-loop system consisting of a high-performance skin with multichannel tactile sensors, a data acquisition/interpretation chip, and a feedback control that operates at a rate of approximately 0.017 kPa^−1^. The IITS is a closed-loop system consisting of a high-performance skin with multi-channel tactile sensors, a data acquisition/interpretation chip, and a feedback control that reads the pressure distribution on the manipulator with a sensitivity of approximately 0.017 kPa^−1^ and a wide detection range of up to 250 kPa. Data acquisition and information processing are performed simultaneously during the dynamic gripping process of the manipulator. Finally, as soon as the pressure threshold is exceeded, i.e., if the value is exceeded in a way that could damage the object, the IITS triggers a stop-feedback control command to terminate the gripping process. This robotic hand is expected to be a future prosthesis for assisting the disabled in grasping.

## 6. Challenges and Prospects

This review comprehensively discusses the characteristics, functionalities, sensing modes, energy harvesting, and signal transmission mechanisms of electronic skins (e-skins) while evaluating their applications in health monitoring and personalized medicine. Despite recent advancements, key challenges remain, including efficient energy storage for prolonged use, stable high-throughput signal transmission, and adaptability for specific populations. Addressing these issues requires developing advanced energy harvesting technologies, enhancing real-time data processing and secure wireless transmission, improving the mechanical properties of flexible materials to withstand deformation, and mitigating signal interference in multimodal sensing systems. These undoubtedly increase the initial cost of integrating e-skin into healthcare systems in terms of materials and technology development. But in the long term, e-skin can reduce medical costs, improve the quality of supervision and care, and improve patient health, potentially bringing significant economic benefits to the medical field. Furthermore, expanding e-skin applications to underrepresented groups is critical. Future progress hinges on interdisciplinary innovations in material science, artificial intelligence, and energy technology, enabling e-skins to extend beyond healthcare into human–machine interaction, rehabilitation, and smart living.

## Figures and Tables

**Figure 1 biomedicines-12-02307-f001:**
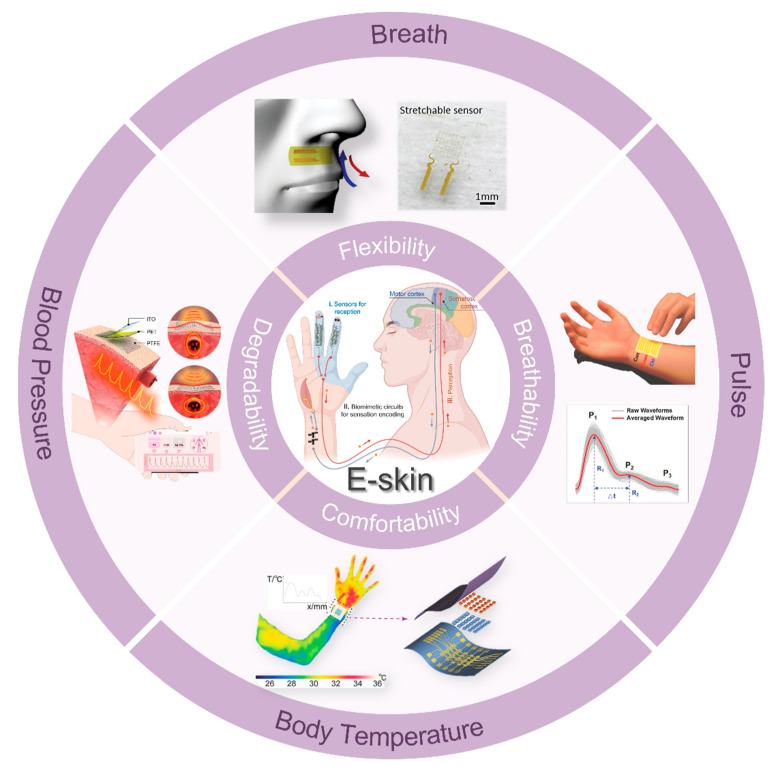
Overview diagram of e-skins with applications involving the monitoring of vital signs such as breath, pulse, blood pressure, and body temperature. Copyright 2024, John Wiley and Sons; Copyright 2022, John Wiley and Sons; Copyright 2020, John Wiley and Sons; Copyright 2018, John Wiley and Sons; E-skin209. Copyright 2023, American Association for the Advancement of Science.

**Figure 2 biomedicines-12-02307-f002:**
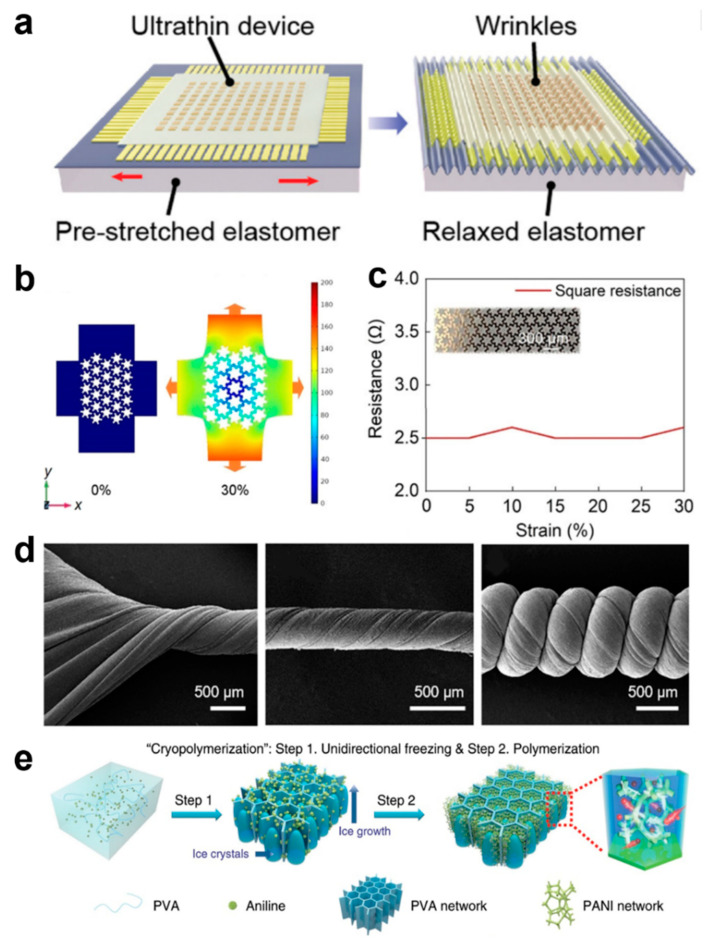
**Pre-Stretching and Mechanical Simulation of Serpentine–Honeycomb Structures for E-Skin.** (**a**) Pre-stretching of device substrates [31]. Copyright 2021, Advanced Materials; (**b**) Mechanical Simulation of Serpentine–Honeycomb Structures [39]. Copyright 2022, John Wiley and Sons; (**c**) Impedance curves for serpentine–honeycomb structures [39]. Copyright 2022, John Wiley and Sons; (**d**) Spiral carbon nanotubes/PU yarns [41]. Copyright 2020, American Chemical Society; (**e**) PVA hydrogel framework with 3D ordered honeycomb structure [41]. Copyright 2020, Springer Nature.

**Figure 4 biomedicines-12-02307-f004:**
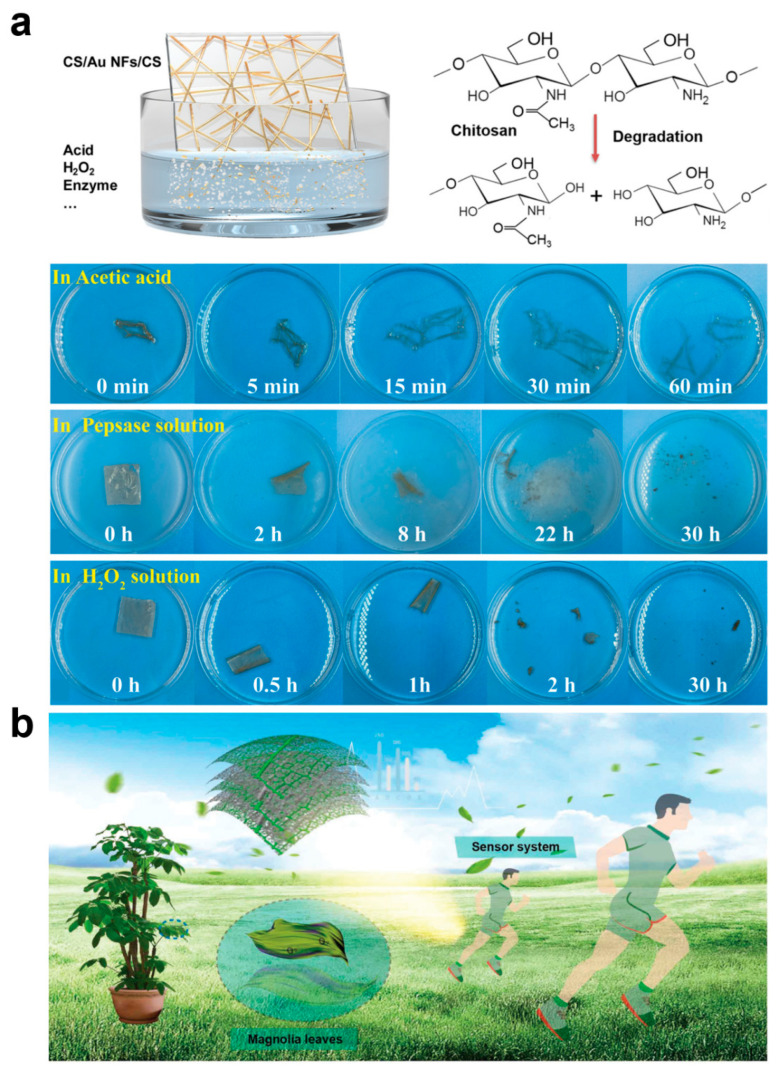
**Biodegradability of Chitosan-Based and Leaf Vein-Derived E-Skin.** (**a**) Decomposition process of chitosan-based e-skin [68]. Copyright 2022, John Wiley and Sons; (**b**) Fully biodegradable e-skin made of natural leaf veins, PLGA/PVA nanofiber film [76]. Copyright 2021, John Wiley and Sons.

**Figure 5 biomedicines-12-02307-f005:**
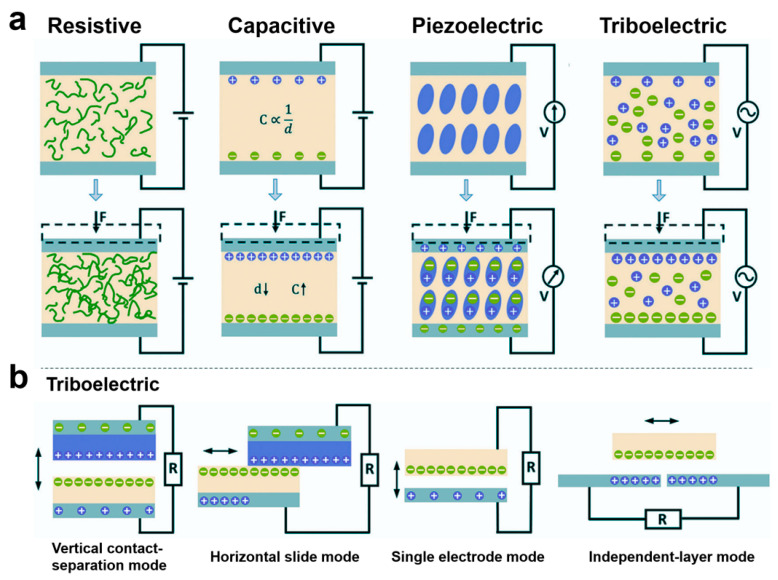
**Sensing Mechanisms and Modes of Operation for E-Skin Sensors.** (**a**) Four sensing modes for pressure sensors. (**b**) Triboelectric e-skins typically have four operating modes.

**Figure 6 biomedicines-12-02307-f006:**
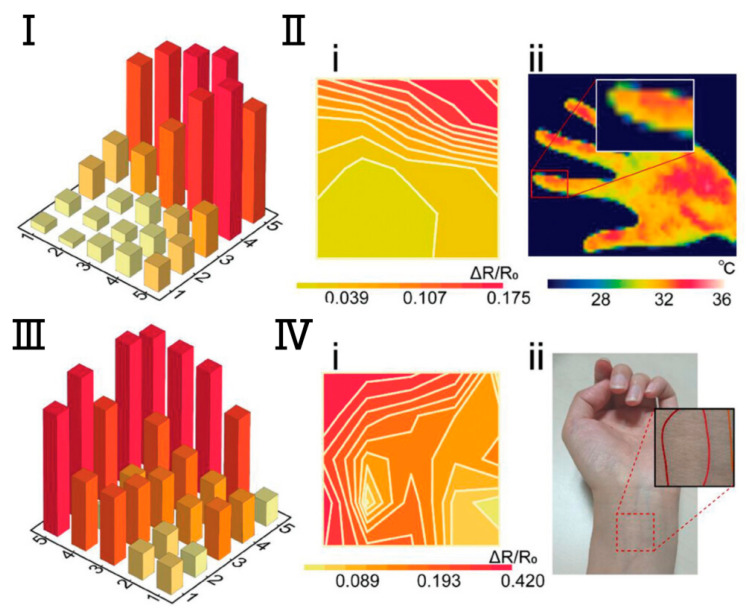
**Temperature Sensing and Recognition with Ionic Gel Resistive E-Skin.** (**I**). Measurement histogram of the detected index finger temperature. The array was put slightly on the finger, and the test area was mainly in (1, 4), (1, 5), (5, 5), and (5, 4) in the pixel matrix. (**II**). Comparison between the measurement results (**i**) and the finger temperature that was obtained by a commercial thermal imager (**ii**). (**III**). Measurement histogram of the detected left wrist temperature. (**Ⅳ**). Comparison between the measurement results (**i**) and the wrist vascular pathways (**ii**) [111]. Copyright 2024, John Wiley and Sons.

**Figure 7 biomedicines-12-02307-f007:**
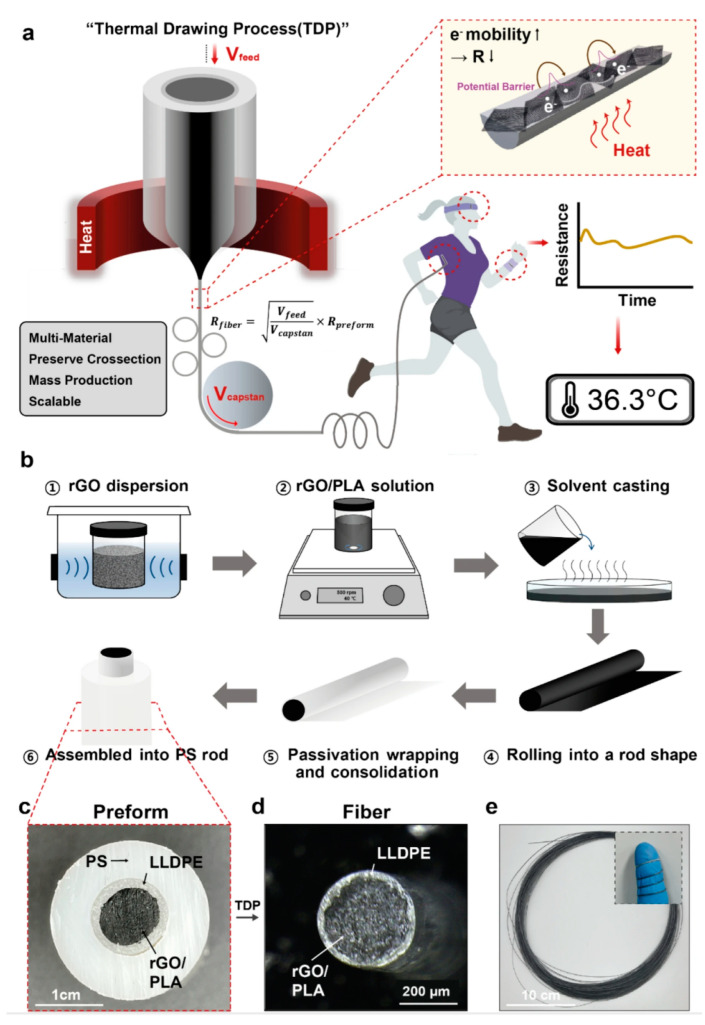
**Fabrication of the thermally drawn, polymer–nanocomposite–fiber temperature sensor** [127]. Copyright 2023, Springer Nature. (**a**) Schematic illustration of the fiber drawing process and the temperature-sensing mechanism of the sensor. (**b**) Fabrication steps of the multi-material preform, including a conductive rGO/PLA temperature-sensing core, an LLDPE passivation layer, and a sacrificial PS cladding. (**c**) Cross-sectional photograph of the multi-layer performance before the TDP. (**d**) Cross-sectional optical microscopic image of the fiber temperature sensor after the thermal drawing and etching processes. (**e**) Photograph of a bundle of fiber temperature sensors before PS etching (inset image: fiber flexibility).

**Figure 8 biomedicines-12-02307-f008:**
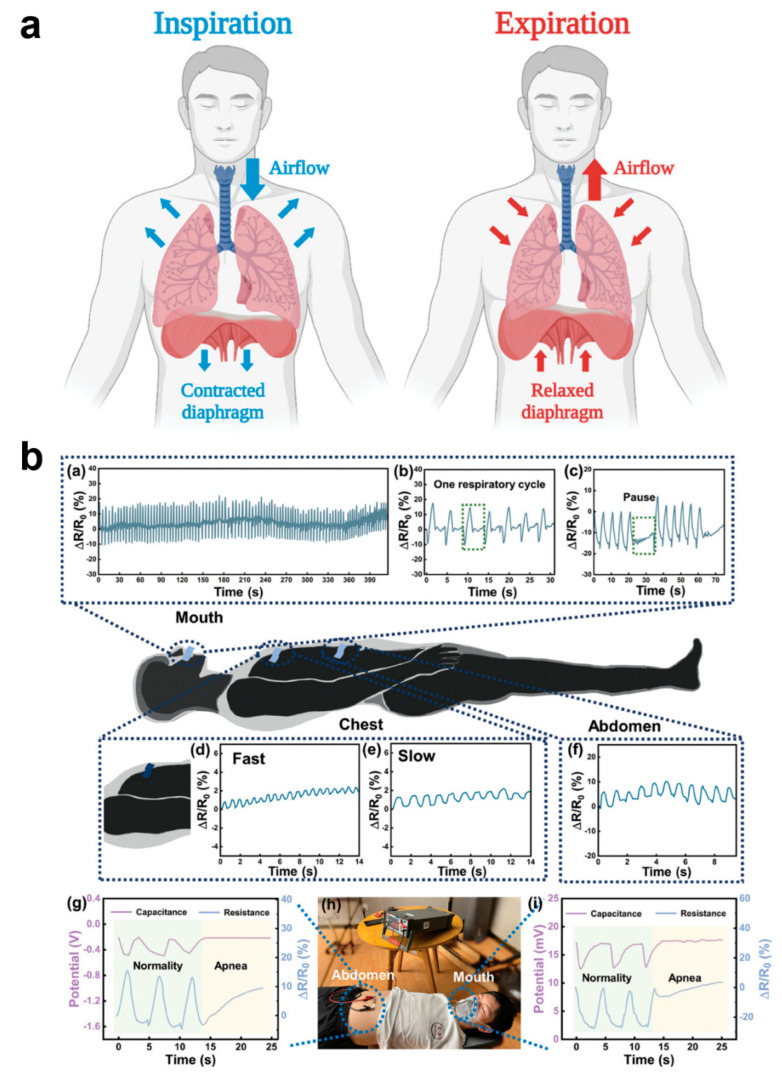
**Wearable E-Skin for Respiratory Monitoring and Pattern Recognition.** (**a**) Schematic illustration of the inspiration and expiration stages of the respiratory cycle [128]. Copyright 2024, John Wiley and Sons; (**b**) A wearable e-skin for monitoring human respiratory rate for measuring respiratory flow, frequency, and recognizing respiratory patterns [131]. In the upper part (**a**–**c**) are mouth breathing signals in human with normal sleep breathing processes and patient with obstructive sleep apnea syndrome. In the middle (**d**–**f**) are the chest signals during fast and slow breathing and the resistance signals in the abdomen during breathing. In the lower part (**g**–**i**) are electrical signals from abdominal and oral airflow activity. Copyright 2024, John Wiley and Sons.

**Figure 9 biomedicines-12-02307-f009:**
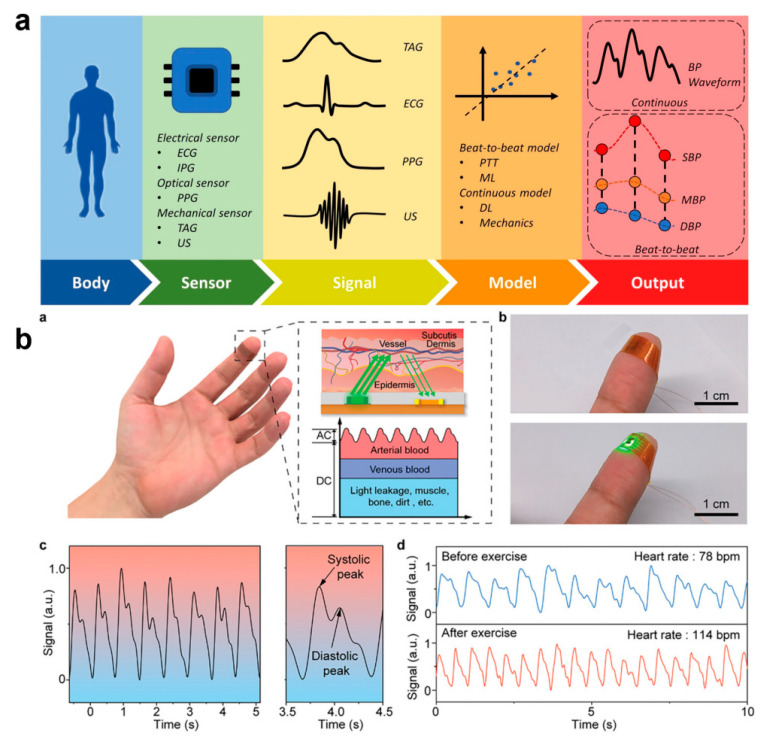
**Cuffless Pulse and Blood Pressure Monitoring with E-Skin Technology.** (**a**) Components of a wearable pulse/blood pressure monitoring device [136]. Copyright 2023, Springer Nature; (**b**) Wearable pulse monitoring sensor based on perovskite [143]. (**a**) The working principle diagram of the perovskite photodetector-based PPG sensor. (**b**) Photograph of the PPG sensor in the nonworking (top) and working (bottom) conditions. (**c**) Pulse signal tested by the PPG sensor in 5 s (left) and its partial enlargement (right). (**d**) PPG signals before and after exercise. Copyright 2024, John Wiley and Sons.

**Figure 10 biomedicines-12-02307-f010:**
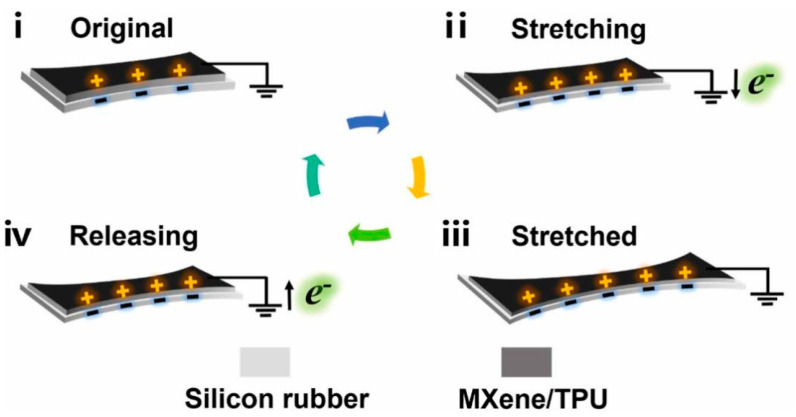
A fifth mode of energy harvesting by TENG that differs from the traditional four—the deformation mode [153]. (**i**) The original state before deformation, (**ii**) both layers are stretched, (**iii**) the deformation is completed, and (**iv**) the stretched film is released gradually. Copyright 2024, Elsevier.

**Figure 11 biomedicines-12-02307-f011:**
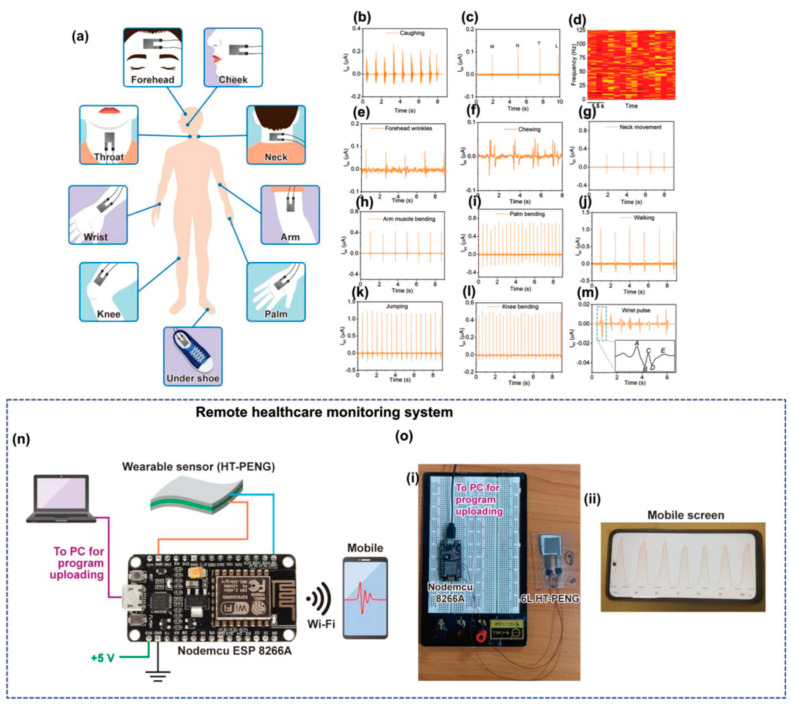
**Real-time application of the hetero-structured-PENG as a self-powered wearable sensor in different parts of the human body for measuring biomedical activity [159]**. Copyright 2023, John Wiley and Sons. (**a**) Human body showing the physiological signal monitoring locations. Output voltage generation from the 6L hetero-structured-PENG used to monitor different human activity: (**b**) vocal cord vibrations: coughing action; (**c**) vibrotactile output signals for different alphabets (M, N, T, and L) and their (**d**) short-term Fourier transform (STFT) spectrogram, (**e**) forehead wrinkling, (**f**) chewing while eating, (**g**) neck movement, (**h**) arm muscle movement, (**i**) palm bending, (**j**) walking, (**k**) jumping, (**l**) knee joint motion, and (**m**) wrist pulse, (**n**) circuit diagram of the IoT-based experimental setup. (**o**) (**i**) Digital image of the practical circuit with (**ii**) a mobile screen showing the received sensor data wirelessly using the Blynk app.

**Figure 12 biomedicines-12-02307-f012:**
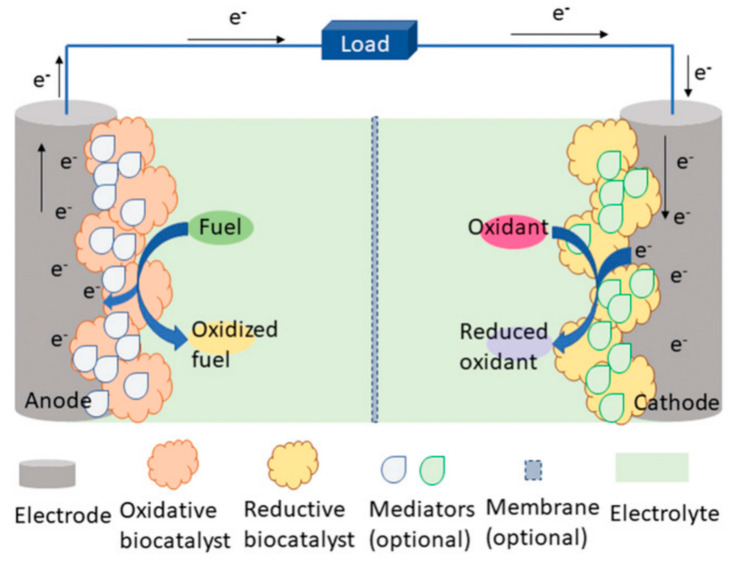
Schematic of a biofuel cell [162]. Copyright 2021, John Wiley and Sons.

**Figure 13 biomedicines-12-02307-f013:**
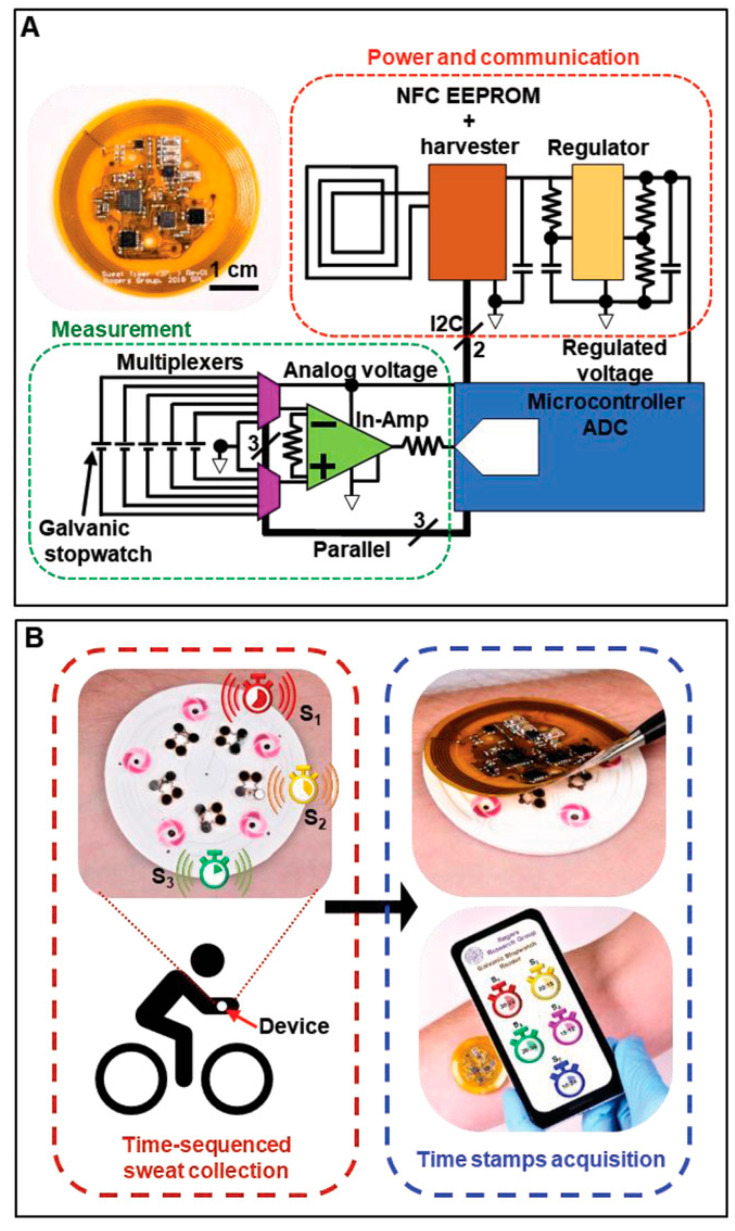
**Flexible NFC System for Wireless Data Transmission in E-Skin** [190]**.** Copyright 2019, John Wiley and Sons. (**A**) Photograph and schematic illustration of a flexible, battery-free NFC electronic system for capturing and wirelessly transmitting voltages on each stopwatch; (**B**) Flowchart delineating standard device usage consisting of on-body sequential sample collection during physical activity (red box) and logging of time stamps using the NFC electronics module (blue box). The smartphone image illustrates a simulated graphical user interface with a suggestive design.

**Figure 14 biomedicines-12-02307-f014:**
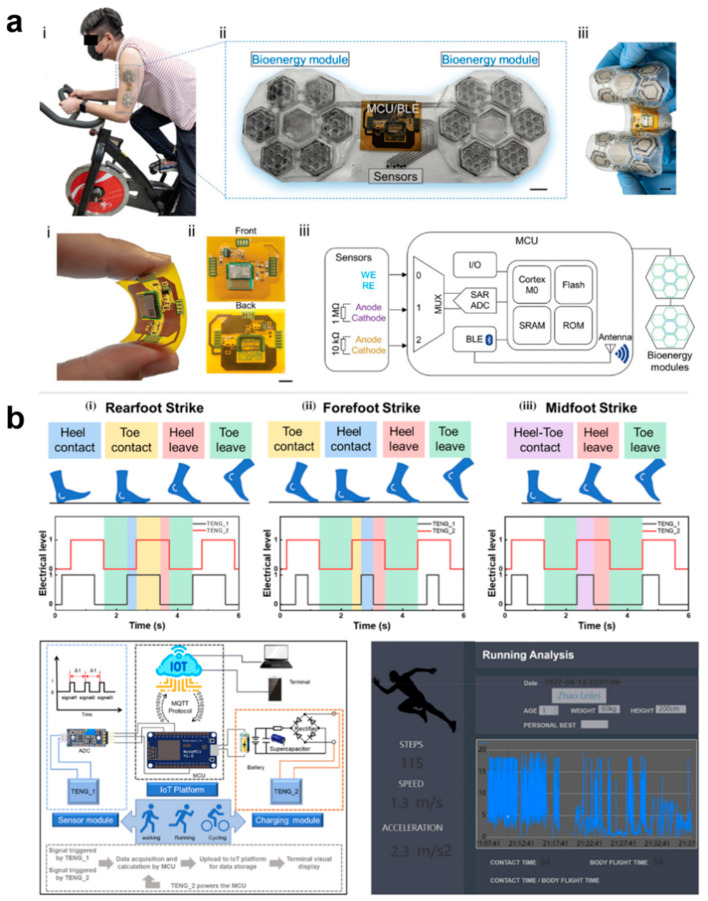
**Self-Powered E-Skin System for Gait Analysis and Motion Detection.** (**a**). Wearable sweat sensing system-based electronic skin for motion detection [191]. At the top, (i) a photo image that the test object wear the integrated E-skin patch, (ii) the composition of the integrated E-skin patch, and (iii) the flexibility of the patch is shown. At the bottom, (i) the FPCB with an integrated MCU under bending deformation, (ii) the front and back side of the FPCB layout, and (iii) schematic diagram of the working principle of the integrated E-skin wearable system. Copyright 2022, John Wiley and Sons; (**b**). The e-skins gait analysis system is used to capture and analyze information such as number of steps, time to touchdown, walking speed, and acceleration [193]. According to the initial landing position of the feet during running, the electrical signals of three different running styles were recorded: (i) rearfoot strike, (ii) forefoot strike and (iii) midfoot strike. Copyright 2024, John Wiley and Sons.

**Figure 15 biomedicines-12-02307-f015:**
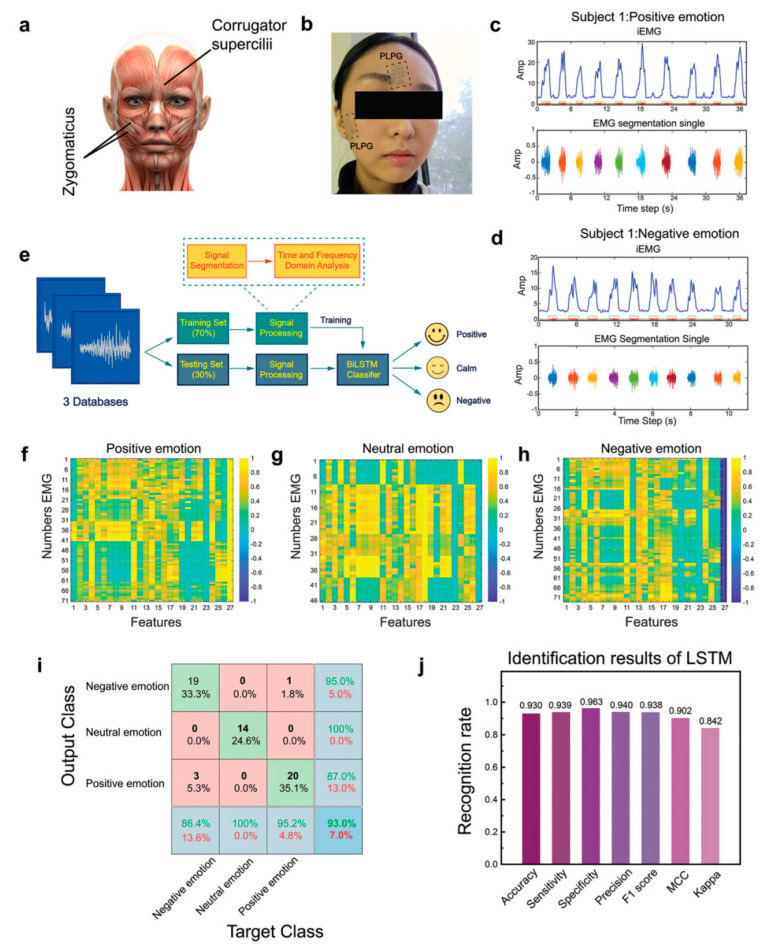
**Facial EMG (fEMG) monitoring by e-skins and machine learning for emotion analysis** [196]**.** Copyright 2024, John Wiley and Sons. (**a**) An image showing the main muscles for emotion expression. (**b**) A photo showing the e-skin with M-3 pattern (denoted by PLPG below) electrodes for fEMG acquisition. (**c**,**d**) Representative fEMG signals and the corresponding extracted integrated EMG (iEMG) from subject 1 under positive (**c**) and negative (**d**) emotions. (**e**) Flow chart of a machine learning algorithm for emotion classification. (**f**–**h**) Thermogram of correlation coefficient of fEMG characteristics in positive (**f**), neutral (**g**), and negative (**h**) emotions. The 27th column is the classification label. (**i**) Confusion matrix of classification training recognition accuracy. (**j**) Identification results.

**Figure 16 biomedicines-12-02307-f016:**
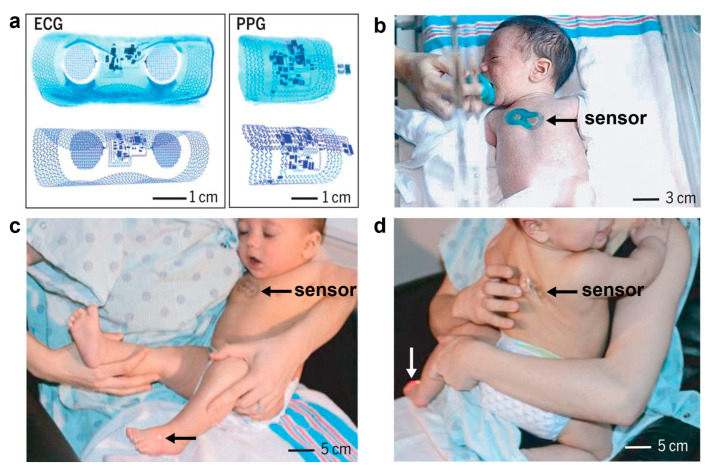
**Wireless, skin-like systems for vital signs monitoring in neonatal intensive care** [207]**.** Copyright 2019, American Association for the Advancement of Science. (**a**). Images and finite-element modeling results for ECG and PPG devices bent around glass cylinders. (**b**). A neonate with an ECG device on the chest. (**c**,**d**). A mother holding her infant with a PPG device on the foot and an ECG device on the chest (**c**) and on the back (**d**).

**Table 1 biomedicines-12-02307-t001:** Comparison of several wireless communication technologies.

Type	Transmission Speed	Band	Power Consumption	Cost	Safety	Range
RFID	106 kbps	1–100 GHz	/	High/Medium	Medium	Several dozen meters
NFC	424 kbps	13.56 MHz	10 mA	Low	Extremely high	0~10 cm
Bluetooth	1~3 Mbps	2.4 GHz	30 mA	Medium	High	100 m
BLE	1 Mbps	2.4 GHz	15 mA	Medium	High	>100 m

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
