# Peer review of "E-Skin and Its Advanced Applications in Ubiquitous Health Monitoring"

_biomedicines, 2024, doi:10.3390/biomedicines12102307_

Round 1

Reviewer 1 Report

Comments and Suggestions for Authors

Manuscript entitled “E-skin and its advanced applications in ubiquitous health monitoring” by “Sun et al” deliberated on the characteristics, fundamentals, new principles, key technologies and their specific applications in health management, exercise monitoring, emotion and heart monitoring, etc. that advanced e-skin needs to have in the healthcare field. Significance in infant and childcare, elderly care, and assistive devices for the disabled is analyzed. Authors focused on an interesting topic; however, the following changes need to be made before its acceptance.

1.      The manuscript has a lot of information, however there are some lacking connectors, or the writing style makes it very confusing. I suggest authors take a closer look and adjust the write up to be more precise and appealing to the readers.

2.      More discussion on the energy harvesting based electronic skin will enhance clarity, discuss these; A biocompatible triboelectric nanogenerator-based edible electronic skin for morse code transmitters and smart healthcare applications; Revolutionizing waste-to-energy: harnessing the power of triboelectric nanogenerators.

3.      Suggest specific directions for future research based on your findings. What aspects remain unexplored or merit further investigation?

4.      Clearly state what unique insights or findings your review brings to the field. For instance, mention if this is the first comprehensive review of its kind or if it synthesizes data in a novel way.

5.      Please strengthen the conclusion by summarizing the key findings, their implications, and potential future research directions clearly and concisely.

6.      In Figure 12 authors kept schematic of a biofuel cell, please mention its significance in relation to the current study.

Comments on the Quality of English Language

Minor editing of English language required.

Author Response

Comment 1: The manuscript has a lot of information, however there are some lacking connectors, or the writing style makes it very confusing. I suggest authors take a closer look and adjust the write up to be more precise and appealing to the readers.

Response 1: Thank you very much for your time during the review process. Your valuable suggestions are much appreciated. We have revised and optimized the semantically incoherent sentences in the article and marked them in red in the manuscript

Comment 2: More discussion on the energy harvesting based electronic skin will enhance clarity, discuss these; A biocompatible triboelectric nanogenerator-based edible electronic skin for morse code transmitters and smart healthcare applications; Revolutionizing waste-to-energy: harnessing the power of triboelectric nanogenerators.

Response 2: Thank you for your valuable suggestions. We have added a discussion of article “A biocompatible triboelectric nanogenerator-based edible electronic skin for morse code transmitters and smart healthcare applications; Revolutionizing waste-to-energy: harnessing the power of triboelectric nanogenerators.” and referenced it.

Comment 3: Suggest specific directions for future research based on your findings. What aspects remain unexplored or merit further investigation?

Response 3: Thanks to your comments, we believe that the future electronic skin needs more exploration and development in three areas: the stability of stretchable electrodes in the working state, the calibration of the electronic skin under different metrics, and high-toughness materials. Additions on these issues are likewise added to the manuscript.

Comment 4: Clearly state what unique insights or findings your review brings to the field. For instance, mention if this is the first comprehensive review of its kind or if it synthesizes data in a novel way.

Response 4: Thank you for your valuable comment. Our article is the first review of its kind that integrates the sensing principles, technical components and health applications of e-skin. The field of e-skin is rapidly evolving, with advanced and novel sensing methods, fabrication techniques, energy storage technologies, health applications, and more being developed every year, and our review provides a comprehensive overview of the evolution of e-skin in health and state-of-the-art work.

Comment 5: Please strengthen the conclusion by summarizing the key findings, their implications, and potential future research directions clearly and concisely.

Response 5: Thank you very much for your comments, and we have strengthened the conclusions section of the manuscript to succinctly summarize the main findings, their implications, and possible directions for future research.

Comment 6: In Figure 12 authors kept schematic of a biofuel cell, please mention its significance in relation to the current study.

Response 6: Thank you very much for your comments, in this part of our work we have introduced the current advanced energy harvesting technology, here energy harvesting by collecting some biological excretions from the human body. In Fig. 12, the schematic of the biofuel cell not only provides a clear and concise schematic for readers who are not familiar with the field, but also provides some inspiration and ideas for advanced e-skin energy storage technologies.

Reviewer 2 Report

Comments and Suggestions for Authors

The manuscript ID "biomedicines-3209380 " having the title "        E-skin and its advanced applications in ubiquitous health monitoring" has summarized the studies related to the integration of E-skin as a health monitoring tool. There are some issues with the current version of the manuscript, therefore, in my opinion it should be revised before publication. Please add some lines considering my observations.

The authors acknowledge the limitations of current energy storage technologies for continuous e-skin monitoring but do not propose new energy gathering methods, relying on self-powered technologies. In addition to these, authors acknowledge the difficulties of efficiently and accurately handling large amounts of data, but do not propose real-time processing methods. Also, authors acknowledge the need for flexible and tough materials but do not explore more advanced materials beyond PDMS and SEBS in this review. I have seen some contexts where authors discussed e-skins for infants and the elderly, but their customizations and requirements are not thoroughly explained. Sometimes authors discuss data transmission and stability for e-skins, but do not mention any concern about the data breaches or the ethical management of sensitive health information during wireless transmissions. Also, authors briefly address wearability concerns but do not provide long-term studies or evaluations of e-skins. Also, in my opinion cost is a major factor for practical application of the e-skin, authors should mention tentative costs of integrating these e-skins into healthcare systems and it may hinder their widespread adoption.

Some editing comments on this paper are as follow:

1.      Authors should incorporate a graphical abstract highlighting the importance of the E Skin. Overview diagram can be used as graphical abstract, please rename it as graphical abstract.

2.      I would strongly suggest writing “Reproduced after taking copyright clearances from the reference” for figures used in this draft. Please take the necessary copyright clearances.

Comments on the Quality of English Language

I have seen many syntax errors; there are errors in punctuation marks such as comma in the whole manuscript. Please check for grammatical errors too.

“Convenient way of life. Convenient way of life" in section 6 should only appear once.

In "As technology has shown a burst of progress, the fourth industrial revolution, which is dominated by artificial intelligence technology, digitalization technology, and the Internet of Things (IoT), has erupted and." The sentence ends abruptly with "and." Consider removing the "and" or completing the sentence.

Author Response

Comment: The authors acknowledge the limitations of current energy storage technologies for continuous e-skin monitoring but do not propose new energy gathering methods, relying on self-powered technologies. In addition to these, authors acknowledge the difficulties of efficiently and accurately handling large amounts of data, but do not propose real-time processing methods. Also, authors acknowledge the need for flexible and tough materials but do not explore more advanced materials beyond PDMS and SEBS in this review. I have seen some contexts where authors discussed e-skins for infants and the elderly, but their customizations and requirements are not thoroughly explained. Sometimes authors discuss data transmission and stability for e-skins, but do not mention any concern about the data breaches or the ethical management of sensitive health information during wireless transmissions. Also, authors briefly address wearability concerns but do not provide long-term studies or evaluations of e-skins. Also, in my opinion cost is a major factor for practical application of the e-skin, authors should mention tentative costs of integrating these e-skins into healthcare systems and it may hinder their widespread adoption.

Response: Thank you very much for your comments. Current energy storage technologies limit continuous e-skin detection; however, the most advanced methods currently available, in addition to adding additional power sources, are the wearing of e-skins with self-powered technology and the development of supercapacitors for energy storage, both of which are described in the manuscript. In addition, efficient and accurate processing of large amounts of data is very difficult in terms of detection, extraction, transmission, and real-time analysis, and the processing methods may need to rely on the development of a full range of devices, for example, cross-electrodes instead of planar electrodes in the structural design of the device, high-quality and high-throughput data transmission ports in data transmission, and the development of more efficient and convenient algorithms for data processing, etc. These elements increase the number of data processing methods. algorithms, etc., which are added to the manuscript. Flexible and tough materials have been discussed in the paper, the traditional PDMS and SEBS do not have this property, at present, PVA ionogel prepared by freeze-thawing method can regulate the Young's modulus, thus having flexible and tough properties. Concerning e-skin for infants and the elderly add corresponding discussions and explanations in the manuscript. I am not sure about the ethical management of sensitive health information during data leakage or wireless transmission, so I did not discuss whether it is possible to minimize this situation by improving the security of data transmission, which is compared in the manuscript. The conclusion section is strengthened at the end of the manuscript to succinctly summarize the main findings, their implications, and possible directions for future research. Finally, cost is indeed a major factor in the practical application of e-skins, and since the issue of cost is not given in many of the referenced articles, and therefore the issue of cost cannot be systematically and extensively investigated, we have added a section in the manuscript that discusses the cost and adds a view of the initial cost of integrating these e-skins into a healthcare system.

Comment 1:Authors should incorporate a graphical abstract highlighting the importance of the E Skin. Overview diagram can be used as graphical abstract, please rename it as graphical abstract.

Response 1: Thank you very much for your time involved in reviewing the manuscript. I have renamed Figure 1 as a graphical abstract.

Comment 2:I would strongly suggest writing “Reproduced after taking copyright clearances from the reference” for figures used in this draft. Please take the necessary copyright clearances.

Response 2: Thank you for your comment. Copyright clearances have been added to all images and all copyright clearances statements have been submitted to the editor.

Comments on the Quality of English Language: I have seen many syntax errors; there are errors in punctuation marks such as comma in the whole manuscript. Please check for grammatical errors too.

Response: Thank you for your comments. Based on your suggestions, we carefully reviewed the manuscript. The errors you pointed out have been corrected, as well as some other grammatical errors, linguistic logic and formatting issues in the article.

Reviewer 3 Report

Comments and Suggestions for Authors

This work reports the recent progress of e-skins in the aspects of device characteristics, functions, sensing modes, energy harvesting and signal transmission, and their applications in human health monitoring and personalized medicine. Overall it is well written and considered for publication, but the following comments should be addressed first.

1.       It is suggested to add a summary title for each figure such as Figure 2 onward, so that the readers can quickly catch and better understand the idea of the whole figure.

2.       The original marks in the cited figures should be removed, otherwise it may cause confusions. E.g., those in Figure 2d, Figure 4a, etc.

3.       Friction electric is more commonly referred as triboelectric, please unify the terminology, because friction is not always required which can also be a close contact of two materials.

4.       Resolution of Figure 11 is too low.

5.       Some recent publications are suggested to broaden the background coverage of flexible electronics in terms of multimodality and scalability, e.g., ACS nano 2023, 17 (2), 1355-1371, Nature communications 2020, 11 (1), 4609, etc.

6.       The authors should go through the entire manuscript more prudently since there are quite a number of typos and errors, such as citation directly jump from 5 to 212 in Line 70, format of citation in Line 73, “...(Fig. 5a) .” with an extra space in Line 353, etc.

Comments on the Quality of English Language

Minor corrections on the description and formatting are required.

Author Response

Comment 1:It is suggested to add a summary title for each figure such as Figure 2 onward, so that the readers can quickly catch and better understand the idea of the whole figure.

Response 1: Thank you very much for your time involved in reviewing the manuscript. In accordance with your suggestion, we have added a summary title for each figure.

Comment 2:The original marks in the cited figures should be removed, otherwise it may cause confusions. E.g., those in Figure 2d, Figure 4a, etc.

Response 2: Thank you for your comment. We removed some marks in the cited figures.

Comment 3: Friction electric is more commonly referred as triboelectric, please unify the terminology, because friction is not always required which can also be a close contact of two materials.

Response 3: Thank you for your comment. As a result of your suggestion, we have changed some of the non-technical terms in the article and standardized the terminology.

Comment 4: Resolution of Figure 11 is too low.

Response 4: Thank you for your comment. We reproduced the picture from the original document to make it clearer.

Comment 5: Some recent publications are suggested to broaden the background coverage of flexible electronics in terms of multimodality and scalability, e.g., ACS nano 2023, 17 (2), 1355-1371, Nature communications 2020, 11 (1), 4609, etc.

Response 5: Thank you very much for your comments, we have added multimodal and scalability aspects to expand the background coverage of flexible electronics Introduction, carefully studied and cited the following two articles: ACS nano 2023, 17 (2), 1355-1371, Nature communications 2020, 11 (1), 4609

Comment 6: The authors should go through the entire manuscript more prudently since there are quite a number of typos and errors, such as citation directly jump from 5 to 212 in Line 70, format of citation in Line 73, “...(Fig. 5a) .” with an extra space in Line 353, etc.

Response 6: Thank you for your comment. Based on your suggestions, we have carefully examined the manuscript and revised some grammatical errors, linguistic logic and formatting issues in the article.

Round 2

Reviewer 1 Report

Comments and Suggestions for Authors

Accept